# Experimental study of synergistic reinforcement of silty clay with glutinous rice paste and MICP

Qizhi Hu●*, Qian Chen

Hubei University of Technology, Wuhan, China

* 102010778@hbut.edu.cn

**Data Availability Statement:** All relevant data are within the paper and its Supporting Information files.

**Funding:** The authors received no specific funding for this work.

## Abstract

The use of microbially induced calcium carbonate precipitation (MICP) technology can improve the mechanical properties of silty clay, and glutinous rice paste can enhance microbial activity, improve the conversion rate of $CaCO_3$ precipitation, and help increase soil strength. An MICP solidification test of silty clay was carried out by adding different concentrations of aged glutinous rice slurry and cementing liquid, and unconfined compressive strength tests and scanning electron microscope analysis of the solidified samples were carried out. The strength growth mechanism of the glutinous rice paste was investigated, and the results revealed that glutinous rice slurry can improve the enzymatic activity of microorganisms, that is, the microorganisms can produce more urease to decompose urea, and as the amount of urease increases, the concentration of the cementing solution increases, and the calcium carbonate generated by the MICP precipitates. When the concentration of the added cooked glutinous rice slurry was 5%, the unconfined compressive strength of the soil was the largest. In addition, the scanning electron microscope analysis revealed that cooled glutinous rice slurry can be used as a bridge to generate a large amount of ineffective carbonic acid. Calcium atoms are connected together to form effective calcium carbonate, which fills in the pores of the soil as a whole, increasing the compactness of the soil and greatly improving its macroscopic mechanical strength.

## 1 Introduction

Microbially induced carbonate precipitation (MICP) is a new green environmental protection technique for solidifying soil [1]. The essence is that the urease produced by microorganisms hydrolyzes the urea to produce $CO_3^{2-}$, which then combines with the $Ca^{2+}$ in cementing solution to produce $CaCO_3$. The chemical reactions are shown in Eqs (1) and (2). The precipitate cements the soil, changing the structure of the soil, and finally, it improves the strength of the soil. Experiments have shown that the quality of this soil fixation method is related to the amount of calcium carbonate generated, as well as the type and

**Competing interests:** The authors have declared that no competing interests exist.

particle size distribution of solidified soil [2–4].

$$CO(NH_2)_2 + 3H_2O \overset{urease}{\rightarrow} 2NH_4^+ + CO_3^{(2-)}. \tag{1}$$

$$Ca^{2+} + CO_3^{2-} \rightarrow CaCO_3(\downarrow). \tag{2}$$

One of the processes of MICP is the generation of sediment to fill the soil pores. In addition, the number and size of the pores in the soil also affect the final solidification effect of the MICP technique. Jin Guixiao et al. [3] found that the cementation effect of loose soft rock and soil particles with different particle size gradations was significantly different after microbial grouting treatment. Zeng Weihua et al. [4] reported that the solidification effect of sand with a good particle size distribution is more significant than that of sand with a poor particle size distribution, and the cementation effect of sand with a poor particle size distribution is poor because when the soil particle size distribution is poor, the thickness the distribution of the soil particles is poor, and the amount of pores between the soil particles is too great. Therefore, it is extremely important to make better use of external methods to fill the pores between soil particles. In the entire MICP reaction process, the CaCl₂ in the cementing solution is the main source of Ca²⁺. If the soil strength is to be made higher, the concentration of Ca²⁺ should be increased, as well as the concentration of the cementing solution, and finally, more precipitate can be formed. However, this is inconsistent with the current research. According to Yang Yuye [5] and Liu Zhiming [6] the higher the concentration of the cementing fluid is, the better the effect is. When the concentration of the cementing solution is greater than 0.6 mol/L, the curing reaction process of the MICP technique gradually slows down and the strength of the solidified soil even decreases. Ahmed Al Qabany et al. also found that the distribution of the calcium carbonate precipitate produced in the normal MICP technique is better when the concentration of the cementing solution is low [7]. This shows that for the current MICP technique, it is far from sufficient to rely solely on microorganisms. It is necessary to find an additive that can not only act as a filler (assisting the calcium carbonate precipitation to fill the pores) but can also act cooperatively with the microorganisms (improving the utilization rate of the cementing fluid) to improve the MICP technique.

Cured glutinous rice paste was a common building colloid material in ancient China, and its strength is higher after cooling. Yue Jianwei et al. [8, 9] used glutinous rice paste to improve a MICP technique. They found that as the concentration of the glutinous rice paste increased, the strength and internal friction angle of the soil samples initially increased and then decreased, while the cohesion increased linearly, and the glutinous rice paste enhanced the microbial activity. Chen Lindong et al. [10] used cooked glutinous rice paste as a general cementitious material in laterite and found that the cooled glutinous rice paste had a certain strength, so mixing it with laterite improved the engineering mechanical properties of the laterite. Yang Fuwei [11] and Ji Xiaojia [12] studied traditional glutinous rice mortar and found that the concentration of the glutinous rice mortar should be within a certain range. The higher the concentration is, the lower the crystallinity is, the smaller the particles are, and the denser the structure of the calcite is. The cooled glutinous rice paste could bond calcium carbonate nanoparticles well and fill their micropores. In conclusion, the existing research on MICP techniques have made progress in developing methods to improve soil strength, but the mechanism of how to effectively deal with pores in soil and how filling media such as matured glutinous rice paste interact with microorganisms to improve soil strength still needs to be explored and analyzed through experiments.

This test uses the research results involving traditional MICP to solidify the soil, and different amounts of ripened glutinous rice slurry and different concentrations of cement are introduced. The advantage of this research is that it combines the processes of ripened glutinous rice slurry and microorganisms to solidify soil. The cooled ripened glutinous rice slurry itself has a high strength. If it is mixed with soil, the strength of the mixed soil will be significantly improved. Furthermore, the cooked glutinous rice pulp can produce starch branch chains after being decomposed by microorganisms, providing energy for their survival and enabling them to produce more urease. The quality of the MICP research results is fundamentally determined by the amount of urease produced by the microorganisms. Because urease can decompose urea to produce $CO_3^{2-}$, the amount of $CO_3^{2-}$ determines the final amount of $CaCO_3$ generated and ultimately determines the strength value of the soil after solidification. Based on this, through macroscopic strength tests and microscopic electron microscope scanning analysis of the samples after preparation and curing, the mechanism of how the two variables interact to influence the soil solidification via MICP is discussed, providing a theoretical basis for the joint application of additives and MICP technology for soil solidification. In addition, this method reduces the high cost of the MICP process and is expected to be applied in engineering applications. However, in this study, only silty clay was tested, and other soil bodies such as sand and loess have yet to be tested. Whether ripened glutinous rice slurry will react better with other soils cannot be determined from our results. We will continue to conduct experimental research using other soils in a follow-up study.

## 2 Materials and methods

The soil sample was obtained from the project department of the fifth primary school of Guanggu, Hongshan District, Wuhan City, Hubei Province, China. The particle grading curve is shown in Fig 1. The collected soil was sealed in a bag and transported to the laboratory. The measured physical indexes are presented in Table 1. The undisturbed soil was crushed and air-dried indoors. The air-dried soil sample was crushed with a wooden stick on a rubber pad, and the stones in the soil were screened out after crushing (Fig 2). The obtained samples were used to prepare the subsequent soil samples for testing. The optimum moisture content of the soil was determined via the light compaction test, and the optimum moisture content of the selected soil sample was 25%. At this time, the soil sample was in the maximum dry density state, and the soil pores under the maximum dry density did not affect the solidification process of the MICP because the microorganism used in this study was Bacillus, the diameter of which is less than 10 μm. The calcium carbonate crystals generated during curing were less than 5 μm During the MICP process, the calcium carbonate crystals gradually formed calcium carbonate precipitates with a larger diameter by using the Bacillus as the nucleation point until the entire pore was filled.

### 2.1 Bacterial fluid and cementing fluid

The preparation of the bacterial solution and the expansion of the culture are key scientific issues. Pasteurianus was selected as the culture strain in the experiment. The strain (number ATCC11859) was obtained Shanghai Bioresource Collection Center (SHBCC), and the strain was in a freeze-dried powder state when it was received (Fig 3). In order to ensure the microbial activity, a liquid medium was selected for the initial expansion culture, and the pH value of the expansion culture was adjusted using NaOH. The pH value was 7.3 [13], and the bacteria liquid was in an alkaline state. After the preparation was completed, 1 mg of freeze-dried powder was converted into liquid via dissolution in 0.2 ml of water, and then, it was added to the liquid medium, transferred to a constant temperature oscillation incubator, and oscillated for

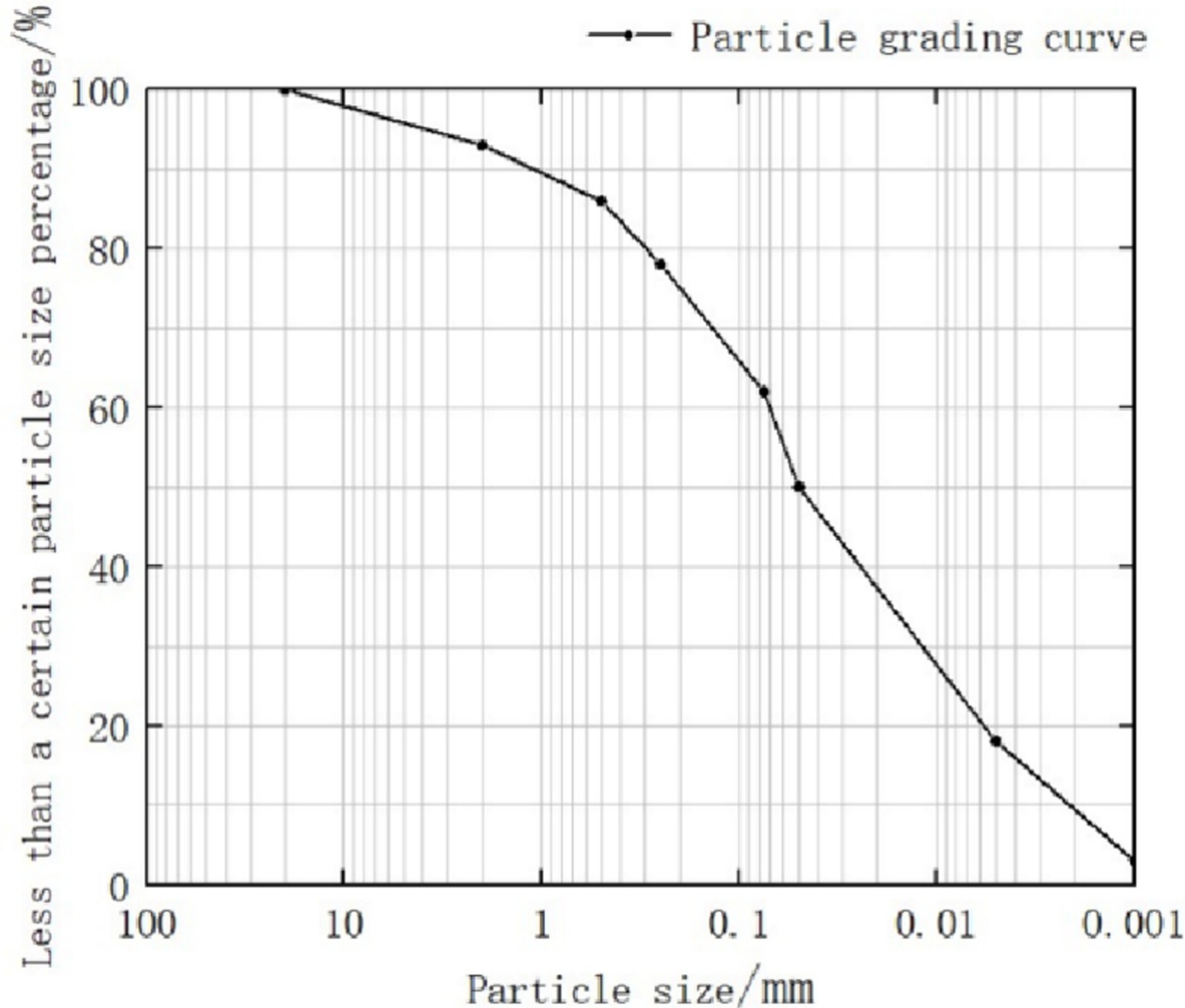

**Fig 1. Grain grading curve.**

36 hours at 30°C and 200 rpm. The above operation was repeated twice, the culture was expanded in an equal ratio, and the concentration of the bacteria liquid was determined after obtaining sufficient bacteria liquid.

**Table 1. Physical properties of the silty clay.**

| Natural moisture content (%) | Void ratio | Liquid limit (%) | Plastic limit (%) | Plasticity index | natural density (KN/m$^3$) |
|---|---|---|---|---|---|
| 28.1 | 0.78 | 34.5 | 19.4 | 15.1 | 19.6 |

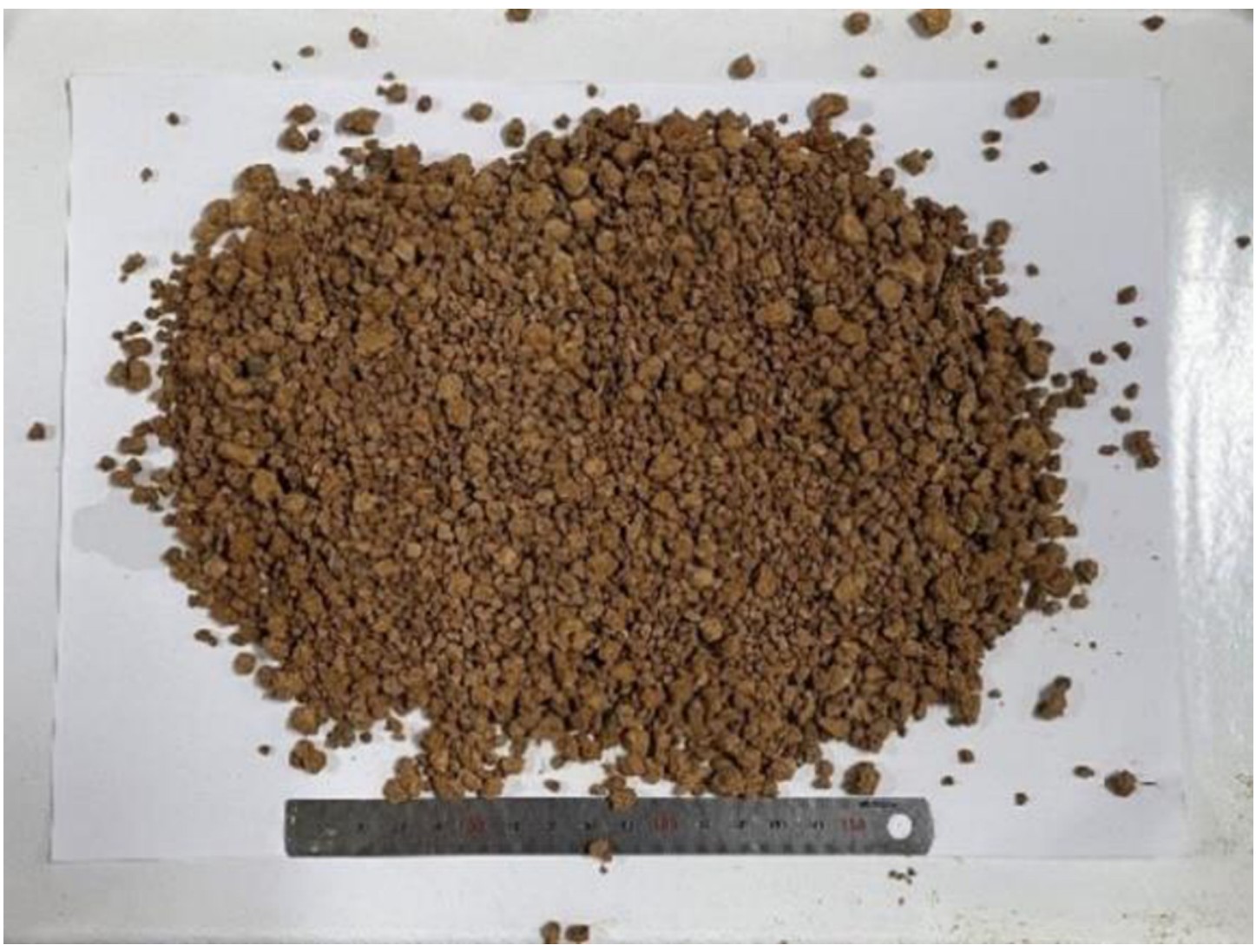

**Fig 2. Air-dried ground soil sample.**

The concentration of the bacteria solution was measured using an ultraviolet-visible spectrophotometer (TU-1810). The concentration of the bacteria liquid was determined by comparing the absorption energy difference between the bacteria liquid and the blank water sample at a wavelength of 600 nm [14], and the optical density at a wavelength of 600 nm ($OD_{600}$) of the test bacteria liquid was finally determined to be 1.32. In order to enable the microorganisms to grow logarithmically (i.e., to achieve the fastest growth rate of the microorganisms [15]), the optical density (OD) was set as 0.1–1. Therefore, in order to ensure the high efficiency of the follow-up test, the bacteria solution was diluted once, and the concentration of the diluted bacteria solution was 0.66, in which the diluent was distilled water of the same quality.

The cementing solution used In the MICP test was a mixed solution of calcium chloride and urea. The calcium chloride provided a source of calcium, the urea provided the $CO_3^{2-}$ for the process in which the microorganisms induced the formation and precipitation of calcium carbonate, and the urea also provided a certain amount of energy for the survival of the

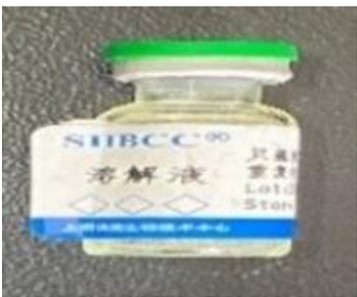

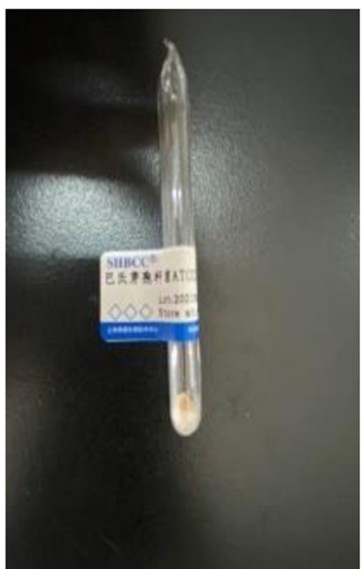

**Fig 3. Microbial freeze-dried powder and bacterial culture solution.**

microorganisms. In order to ensure that the MICP reaction could be fully carried out without additives and the molar mass ratio of the urea to $CaCl_2$ was 1:1, the cementing solution concentrations selected for the experiments were 0.6 mol/L, 0.8 mol/L, and 1.0 mol/L. The method of preparing the 0.6 mol/L cementing solution was as follows: 0.6 mol of urea (36 g) and 0.6 mol of $CaCl_2$ (66 g) were added for every 1 L of distilled water, and the concentration was 0.6 mol/L after stirring evenly. The remaining two concentrations were prepared according to this method for later use.

## 2.2 Glutinous rice paste

The ripened glutinous rice paste used in the experiment was boiled from hydrated glutinous rice paste, which was prepared from glutinous rice flour and water. The glutinous rice flour was purchased from local supermarkets. Hydrated glutinous rice paste with a concentration (mass ratio) of 10% was used in the experiments.

The method of preparing the cooked glutinous rice paste was as follows. First, 100 g of glutinous rice flour were weighed and evenly mixed with 900 g of purified water, and then, the mixture was boiled. In order to ensure that the glutinous rice paste completely matured [9], during the boiling step, it was heated to 90°C while stirring, and the gelatinization time was 40 min. In order to reduce the influence of the evaporation of the water on the concentration of the glutinous rice paste, the water was replenished in a timely manner during the gelatinization to

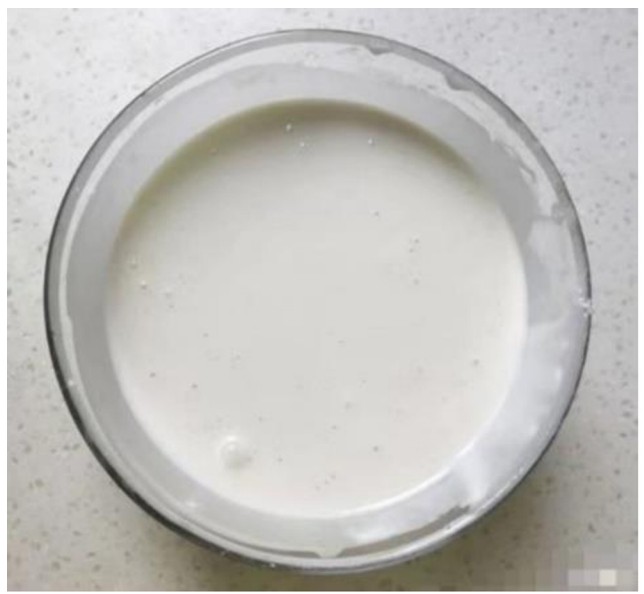
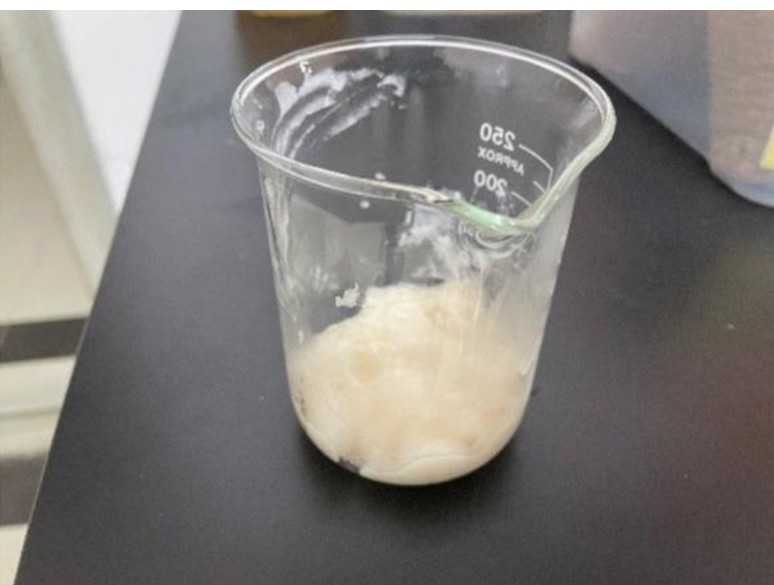

a.

b.

**Fig 4. Cooled glutinous rice paste.** a. Water mixed with glutinous rice flour. b. Boiled and cooled glutinous rice paste.

keep the total quality of pulp unchanged. After the gelatinization, the paste was cooled to room temperature for later use (Fig 4), and at this time, it was characterized by a poor liquidity and a high viscosity.

## 2.3 Methods

**2.3.1 Determination of urease activity.** In this experiment, FE-38 was used to measure the rate at which the conductivity of the bacterial solution+urea solution or the bacterial solution+urea solution+cooked glutinous rice slurry changed to determine whether the addition of cooked glutinous rice slurry caused a change in urease activity. The bacterial solution to be tested and the urea solution were mixed using a magnetic suspension stirrer, and FE-38 was used to measure the change in the conductivity during the first 5 minutes. According to the empirical value obtained by Whiffin, the average change in the conductivity per minute (ms/min) can be converted into the amount of urea hydrolyzed by the urease per unit time, and it can be multiplied by the dilution factor to obtain the amount of urea hydrolyzed by the bacterial solution to be tested per minute. This value is used to represent the urease activity in this paper.

**2.3.2 Unconfined compressive strength test.** A YYW-2 instrument was used to carry out the unconfined compression test on the prepared and cured samples, and to draw the stress-strain curve of each sample.

**2.3.3 Scanning electron microscope analysis.** The fresh surface of the sample after curing was selected for gold spraying, and the morphology, size and distribution of the calcium carbonate induced by the microorganisms were analyzed using scanning electron microscopy (SEM).

## 3 Urease activity and sample preparation

### 3.1 Urease activity assay

The urease activity of the bacteria determines the ability of the microorganisms to decompose the urea. The higher the urease activity is, the stronger the ability of the microorganisms to decompose the urea is. Finally, regarding the macroscopic performance, the more calcium carbonate is generated, so the soil strength is higher. The survival rate of the microorganisms affects the urease activity. The general MICP test does not provide enough energy for the microorganisms to survive. However, cooked glutinous rice paste is composed of amylopectin, which can theoretically provide energy for the microorganisms. The enzyme activity of the bacteria liquid was determined. The purpose of this determination was to ensure that the enzyme activity of the bacteria liquid met the requirements and then to determine whether the cooked glutinous rice paste could improve the enzyme activity of the bacteria liquid.

Two control groups were set up: reaction solution A without glutinous rice paste and reaction solution B with glutinous rice paste (Fig 5). The reaction solution was a mixture of 20 mL of bacteria solution and 180 mL of 1 mol/L urea solution. The change in the conductivity in the first 5 minutes was measured using a conductivity meter (FE-38), which represented the amount of urea hydrolyzed by the bacteria solution per minute, that is, the enzyme activity [16].

According to the measured data, the change in the conductivity of pure bacterial solution B in the first five minutes was 0.0312 mS/(cm · min$^{-1}$). According to the empirical value obtained by Whiffin [17], the average change in conductivity per minute (ms/min) can be converted into the amount of urea hydrolyzed by urease per unit time, which can then be multiplied by

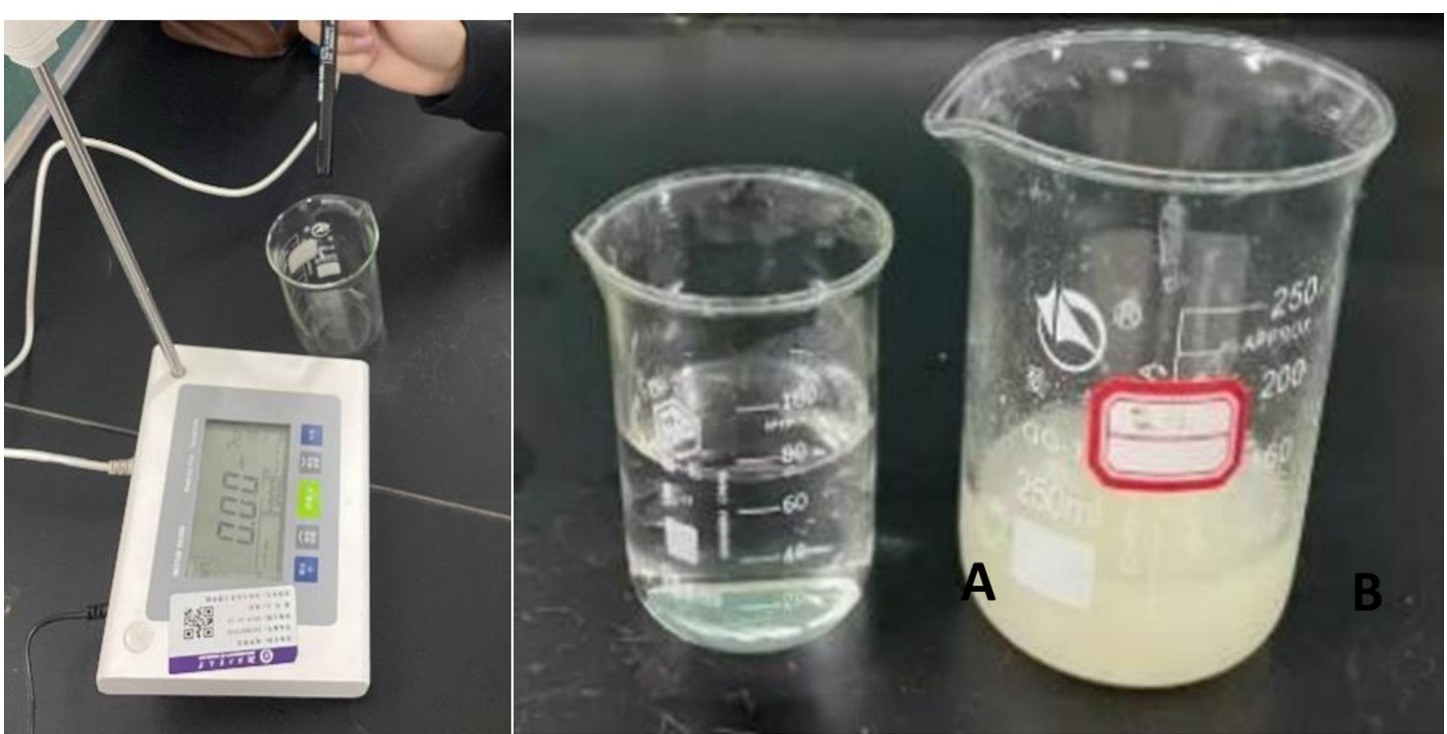

a.                                                                              b.

**Fig 5. Urease activity test.** a. FE-38. b. Reaction liquid to be measured (A, B).

the dilution factor to obtain the amount of urea hydrolyzed per minute by the bacterial solution to be tested. Since the bacterial solution was diluted by 10 times during the enzyme activity test, the average urease activity of the bacterial solution without cooked glutinous rice slurry was 0.312 mS/(cm · min$^{-1}$), and the measured data were plotted to create a urease activity diagram (Fig 6). From the data plotted in the diagram, it can be seen that the urease activity of reaction solution A without glutinous rice slurry added in the early stage was higher than that of reaction solution a with glutinous rice slurry added. The preliminary analysis indicates that the glutinous rice slurry had not yet acted in the early stage, and the bacterial solution in group B restricted the flow of the bacterial solution under the influence of the cooked glutinous rice slurry, and therefore, the enzymatic reaction of microorganisms was inhibited. After about 12 hours, the rate of change of the conductivity of the group B reaction solution became

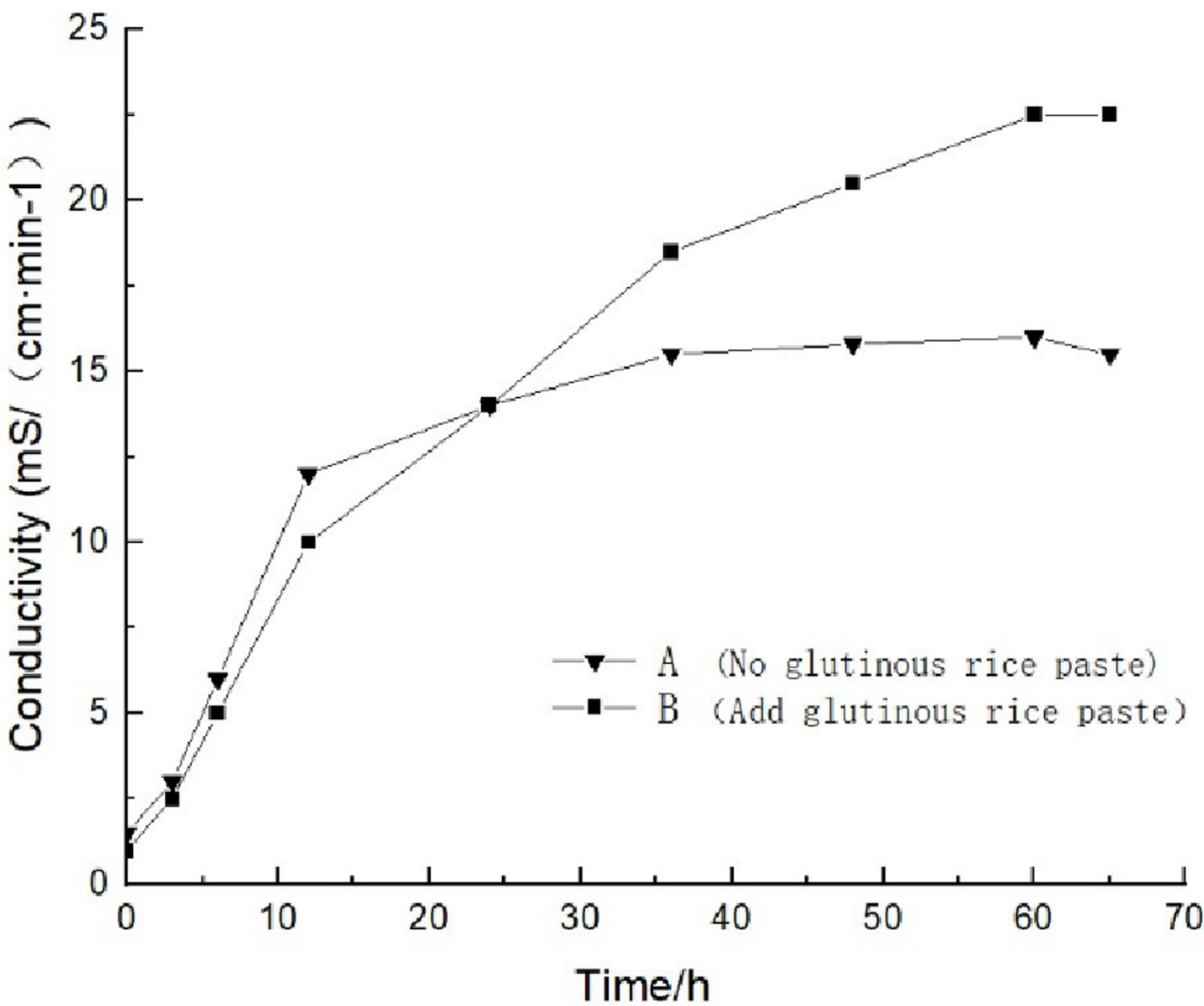

**Fig 6. Plot of conductivity versus time.**

greater than that of the group A reaction solution, and the difference between the two gradually widened, indicating that the cooked glutinous rice slurry began to act at this time and provided energy for the microorganisms. Based on the data obtained from many experiments, it can be concluded that under the proper curing conditions, the urease activity of the bacterial liquid can be increased by adding cooked glutinous rice slurry. However, the practical effect of the glutinous rice slurry in the microbial solidification of the soil needed to be proven via the subsequent macroscopic and microscopic tests.

## 3.2 Test scheme and sample preparation

Sample preparation is the most important scientific problem to be solved during the entire testing process, including the formulation of the testing plan, sample preparation, and sample maintenance. The formulation of the testing plan was mainly based on the three test variables of the study, which are the concentration of the cement, the amount of the cooked glutinous rice paste, and the amount of the bacterial liquid. The purposes of these tests were as follows: 1) to verify the influence of the cement concentration on the MICP process; 2) to verify the effect of the concentration of the cooked glutinous rice slurry on the microbial solidification of soil; and 3) to verify whether the increase in the soil strength is completely caused by the cooked glutinous rice slurry, as well as to set a control group with a bacterial liquid content of 0 ml.

Therefore, the control variable method was used to formulate a testing plan that fit the purposes of the experiment (Table 2).

The mixing method was used in the tests, and a standard sample was prepared according to the optimum moisture content of 25%. According to the optimum moisture content and the mold volume of 80 mm × 39.1 mm, it took about 160 g of plain soil to create the standard sample. Twenty samples needed to be prepared for each group of parallel tests, and 60 samples

**Table 2. Test ratio of each group.**

| Sample No. | Concentration of cementing solution(mol/L) | Content of glutinous rice paste (%)% | Bacterial liquid (ml) Ml |
|---|---|---|---|
| A0 | 0.6 | 0 | 30 |
| B0 | 0.8 | | 30 |
| C0 | 1 | | 30 |
| D0 | 1 | | 0 |
| A1 | 0.6 | 3 | 30 |
| B1 | 0.8 | | 30 |
| C1 | 1 | | 30 |
| D1 | 1 | | 0 |
| A2 | 0.6 | 5 | 30 |
| B2 | 0.8 | | 30 |
| C2 | 1 | | 30 |
| D2 | 1 | | 0 |
| A3 | 0.6 | 7 | 30 |
| B3 | 0.8 | | 30 |
| C3 | 1 | | 30 |
| D3 | 1 | | 0 |
| A4 | 0.6 | 10 | 30 |
| B4 | 0.8 | | 30 |
| C4 | 1 | | 30 |
| D4 | 1 | | 0 |

needed to be prepared for three parallel tests. According to the Standard for Geotechnical Test Methods (GB/T50123-2019), the preparation of a single sample was as follows. First, 160 g of soil were mixed with 30 ml of bacteria liquid (30 ml of distilled water were used instead when the amount of bacteria liquid was 0 ml), and then, the mixture was left to stood for 30 min [18]. The prepared soil containing bacteria was completely mixed with 10 ml of cementing liquid with different concentrations and different doses of cooked glutinous rice paste. After mixing, the soil samples were successively placed in an 80 mm × 39.1 mm mold, and a small amount of Vaseline was added to the inner wall of the mold. The soil samples were divided into four layers, and each layer was compacted 25 times. The quality of the compacted soil in each layer was as consistent as possible. In order to make the contact between the upper and lower layers closer, it was necessary to scratch the soil samples between the layers. After this, it was placed in the curing basin, the mouth of the basin was covered with film, and it was then left to stand for 28 days at an indoor temperature of 25 and a humidity of 65±10 (Fig 7). In Fig 7, the upper label of the sample corresponds to the different variables in the testing plan.

## 4 Analysis of the influence of the curing strength

After the sample was created, the curing test was carried out, and the unconfined compressive strength test was carried out after the curing test. The failure type and peak strength of the test group cured for 28 days were analyzed using a strain-controlled unconfined instrument (YYW-2 type) (Fig 8). The test instrument was hand-operated and unconfined, and the load ring reading and axial deformation could be read and recorded. The strain rate was controlled at 1 mm/min. The axial strain was determined by subtracting the height of the sample from the axial deformation. The axial stress was determined from the load ring reading after a certain coefficient and calibration area. According to the axial strain and axial stress, the peak unconfined compression value of each sample was obtained. At this time, it was necessary to process and fit the data. In this test, Origin software was used to process the data and plot the strain curve.

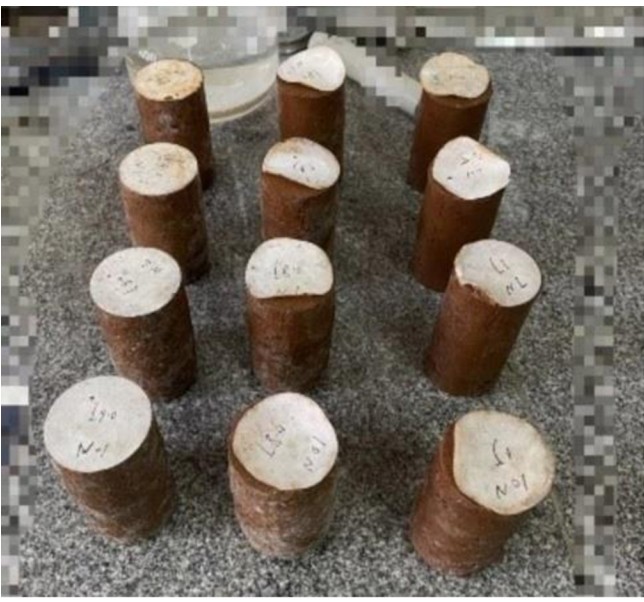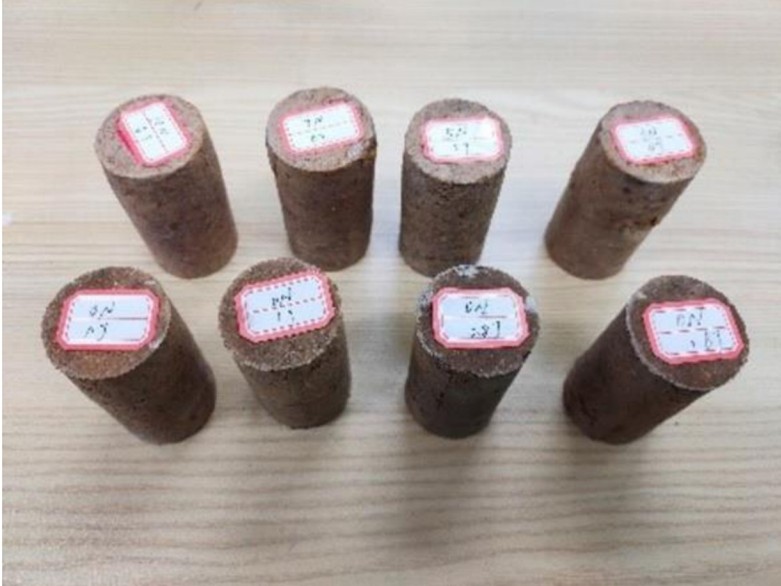

**Fig 7. Soil samples used in the tests.**

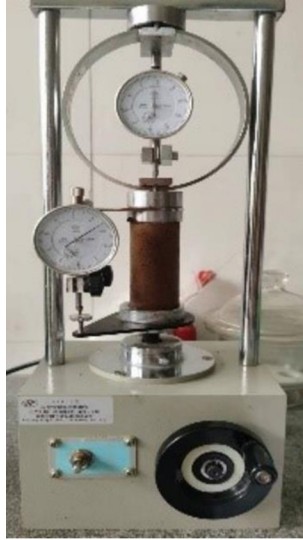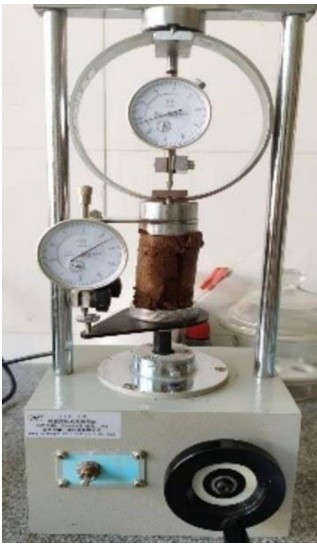

**Fig 8. Unconfined compression test.**

## 4.1 Effect of glutinous rice paste content on soil strength

After curing, the unconfined compression test was carried out on the samples containing different amounts of glutinous rice paste. The strength curve of the samples is shown in Figs 9–12, It can be seen that after adding only cooked glutinous rice paste, the strength of the soil was obviously improved compared with that without adding glutinous rice paste. As is shown in Fig 12, the strength of D0 is only 55.3 kPa. The addition of glutinous rice paste increased the strength by a maximum of 100%, to 113.2 kPa (D4). Under the combined action of microbial solidification and glutinous rice paste, the soil strength generally increased, and in the entire MICP process, the soil strength increased with increasing glutinous rice paste concentration. Fig 9 shows that when the concentration of the cementing solution was 0.6 mol/L, the strength of the soil (A0) treated using the MICP technique reached a maximum of only 70 kPa. After the glutinous rice paste was added, the strength was significantly improved. The strength of the soil samples in the same group was at least 102.6 kPa and at most 143.2 kPa (A4), with an increase of 100%. The other groups of samples with higher cementing fluid concentrations had stronger soil strengths. As can be seen from Figs 9–12, the glutinous rice paste played an extremely important role in enhancing the strength of the soil regardless of whether bacteria liquid filled the soil pores.

The above experimental data show that as the glutinous rice paste concentration increased, the strength of the sample increased. All of the strength curves obtained are similar, indicating that a higher concentration of glutinous rice paste produces a better strengthening effect, and there is a threshold. Below the threshold, the higher the concentration is, the greater the strength of the sample is. After reaching the maximum value (the threshold), the concentration of glutinous rice paste does not affect the strength of the sample. It can be concluded from the data that when the glutinous rice paste content is 5%, the strength value added is almost consistent with those for contents of 7% and 9%. Therefore, from an economic and practical point of view, it can be preliminarily concluded that when the glutinous rice paste content is 5%, the combined effect of the glutinous rice paste and MICP is the best. In addition, it was found that the strength curves of the samples with different amounts of cooked glutinous rice paste almost all exhibited the same growth rate as a previous curve (at this time, the samples were in the

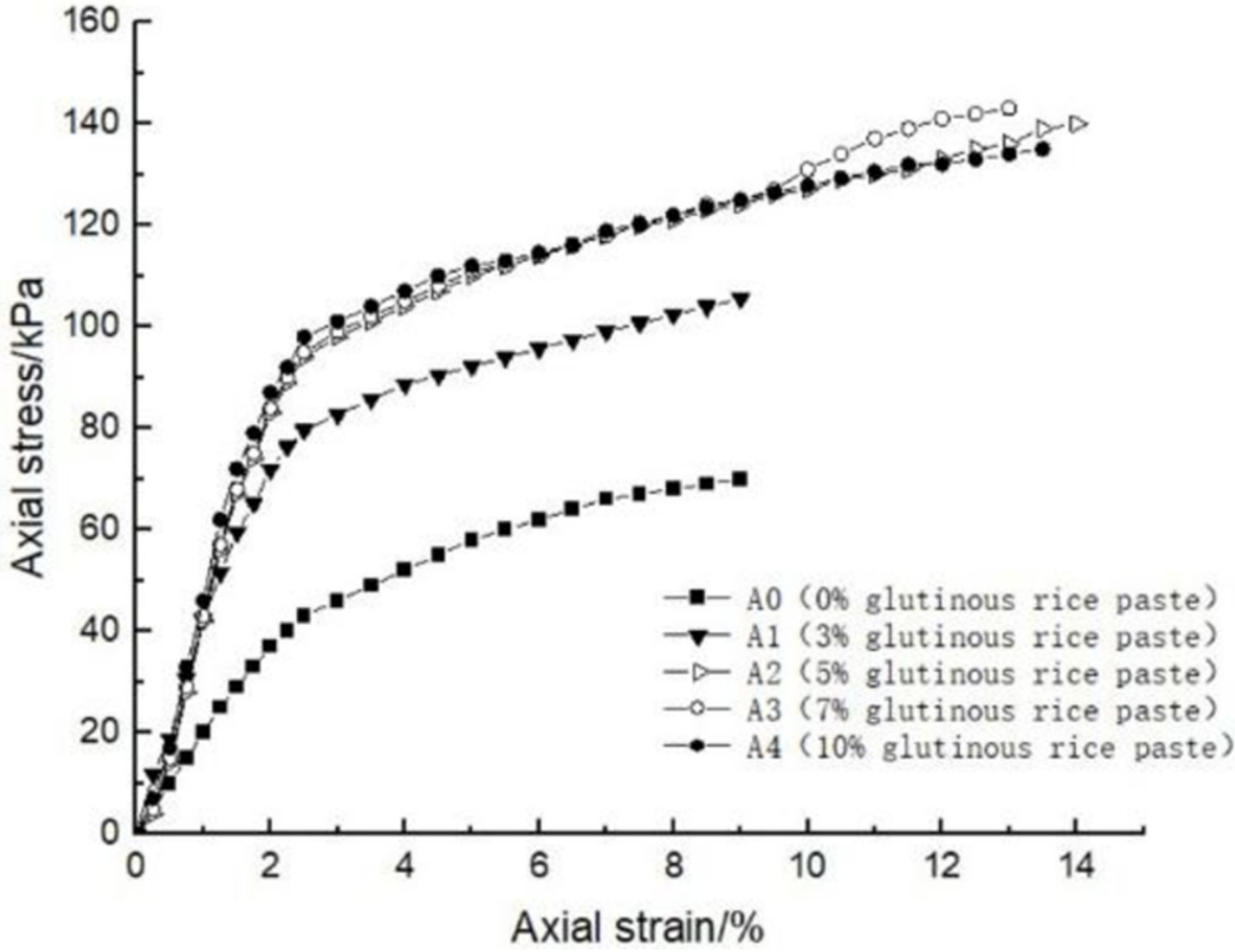

**Fig 9. Strength curve of soil with different glutinous rice paste contents and 0.6 mol/L cementing solution.**

elastic stage). With the gradual increase in stress, the curvature gap in the middle section gradually widened (at this time, the samples were in the elastic-plastic stage), as shown by the test data of group B.As is shown by the data or group B, the reason for this phenomenon is that the strength of the clay itself was lower than that of the calcium carbonate precipitate produced by the reaction and that of the glutinous rice paste after cooling. At the beginning of the unconfined compression test, the clay itself was strong enough to resist the applied load, so the curvatures of the early parts of the curves are roughly the same. However, as the axial pressure increased, the sediment generated in the soil and the cooled glutinous rice paste began to play a role. The soil samples with different dosage of glutinous rice paste and cementing liquid began to exhibit different compressive capacities, so the difference in the rates of the curves gradually widened in the middle section. This also shows that the precipitation generated by the MICP, and the matured glutinous rice paste after cooling played a large role in improving the soil strength.

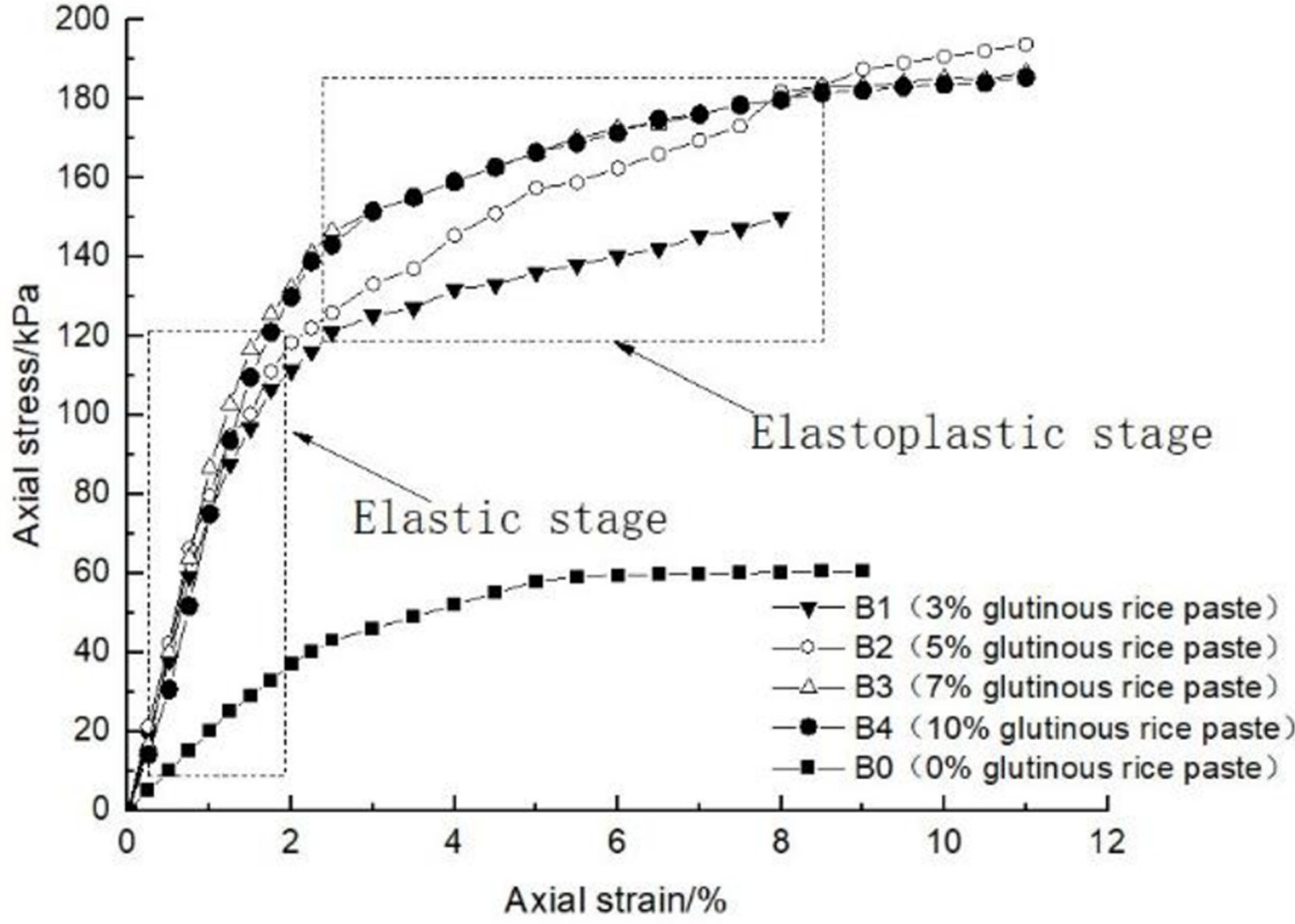

**Fig 10. Strength curve of soil with different glutinous rice paste contents and 0.8 mol/L cementing solution.**

## 4.2 Influence of different cementing fluid concentrations on soil strength

After curing, the unconfined compressive strengths of the samples with different cementing fluid concentrations were tested, and the unconfined compression test data diagram are presented in Figs 13–17. It can be seen that the addition of glutinous rice paste affected the improvement of the strength of the samples caused by the concentration of the cementing solution. Fig 13 shows that as the concentration of the cementing solution gradually increased, the maximum strength of the sample mixed with glutinous rice paste reached 242.5 kPa, while the strength of the sample without glutinous rice paste was only 70 kPa. For the same glutinous rice paste content, the maximum strength of the sample in Group B with 0.8 mol/L cementing solution was 1.3 times that of the sample with 0.6 mol/L cementing solution concentration. For the same content of glutinous rice paste, in Group C, the maximum strength of the sample with 0.6 mol/L cementing solution was twice as high as that of the sample with 1 mol/L cementing solution. All of the experimental data show that the concentration of the cementing solution is an important factor affecting the curing effect of the MICP. According to the test results, under the premise of the addition of cooked glutinous rice paste, the concentration of

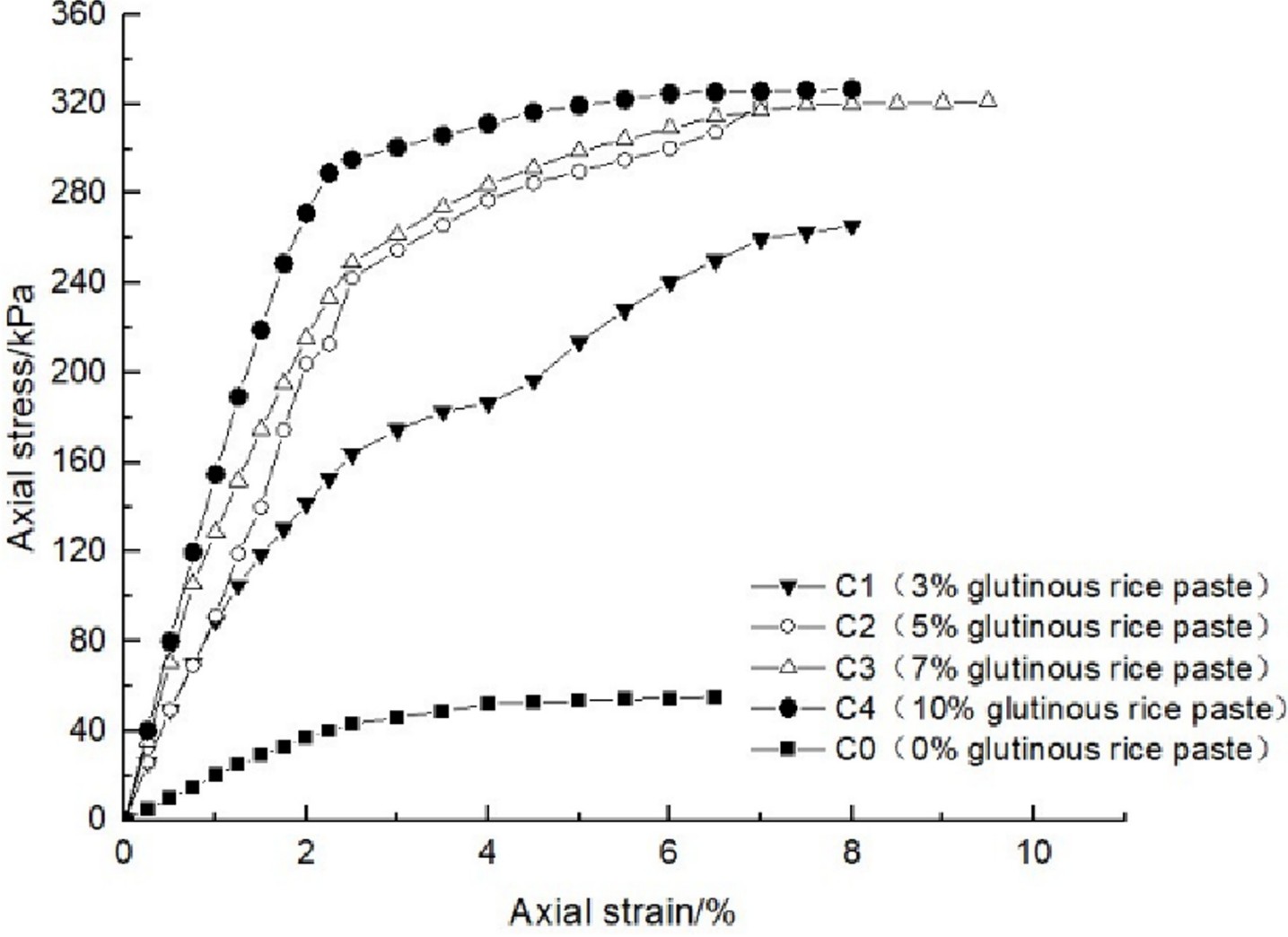

**Fig 11. Strength curve of soil with different glutinous rice paste contents and 1 mol/L cementing solution.**

the cementing solution in the MICP reaction process is within 1 mol/L, and the higher the concentration is, the higher the utilization rate is, which intuitively demonstrated the higher the strength of the sample. Figs 13–17 all show that the strength of the soil without bacteria liquid was lower than that of the soil with bacteria liquid.

In the conventional MICP test, it is usually found that a high concentration of cementing solution will inhibit the microbial reaction and lead to a low soil strength. Many researchers [6, 7] have found that when the concentration of the cementing solution is too high, the strength of the sample will not increase much, and it may even decrease. This is obviously different from the results of our experiments. Our results show that the higher the concentration of the cementing solution is (within 1 mol/L), the higher the strength of the sample is, and the magnitude of the increase is larger. Based on the investigation of urease activity presented in Section 2, it is speculated that the existence of matured glutinous rice paste allows the microorganisms to produce more urease, and the urea can be fully decomposed to maximize the utilization of the cementing solution, thus generating more precipitate. Therefore, the strength of the sample is improved, that is, the existence of cooked glutinous rice paste can improve the

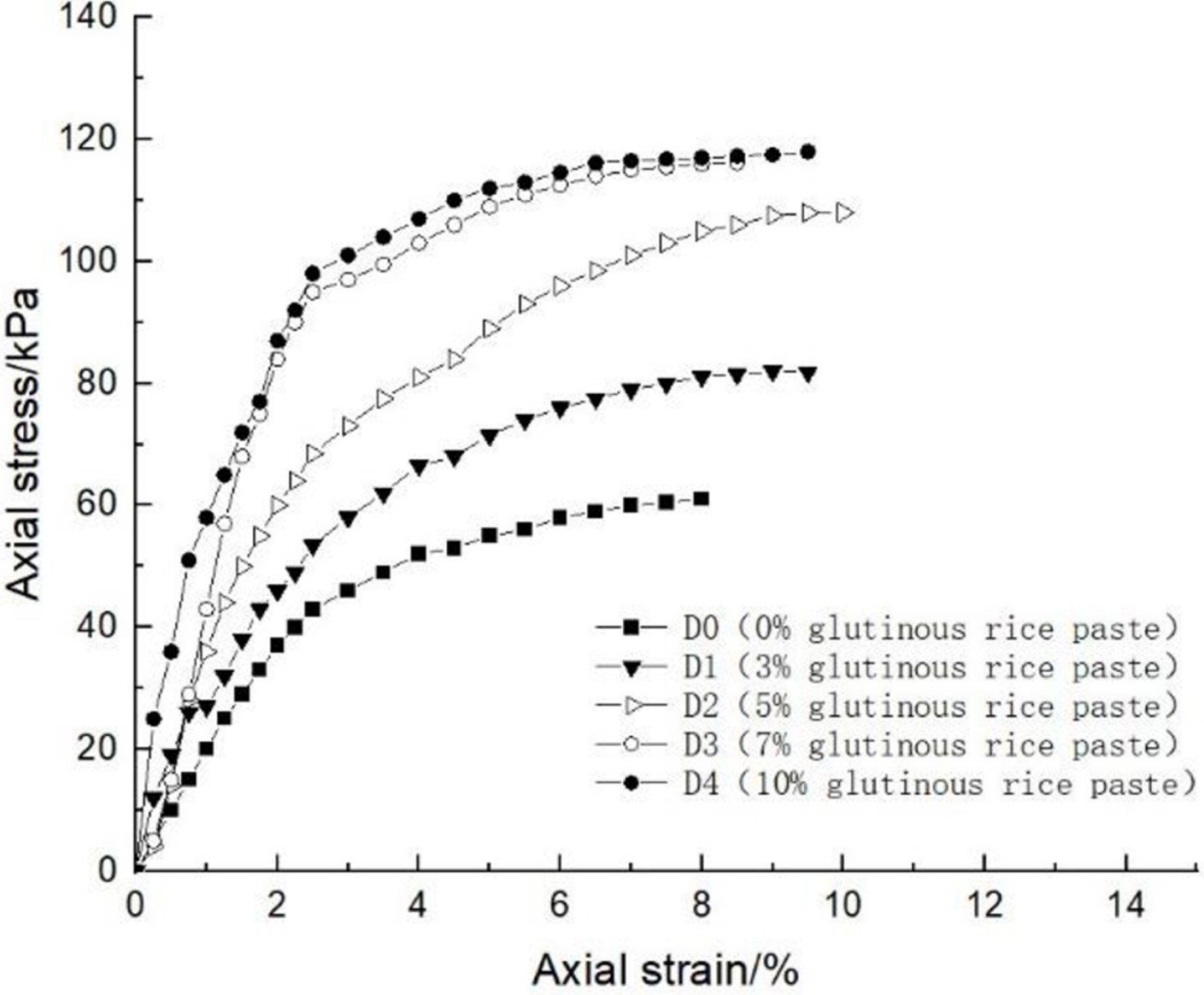

**Fig 12. Strength curve of soil with different glutinous rice paste contents and no cementing solution.**

enzyme activity of the microorganisms. This experimental conclusion is consistent with the research results of Yue Jianwei et al. [8, 9].

## 5 Scanning electron microscope analysis

The results of the unconfined compression test show that the curing effect was best when the content of the glutinous rice slurry was 5% and the concentration of the cement was 0.6 mol/L, so the A0 and A1 samples, as well as the A2, D0, and D2 test soil samples mixed with only matured glutinous rice slurry, were selected for SEM analysis when the content of glutinous rice slurry was 5% and the concentration of the cement was 0.6 mol/L. Before the SEM analysis, the microorganisms and the cured glutinous rice paste were treated together, and then, the cured samples were crushed using an agate mortar. The particle structure of the soil surface

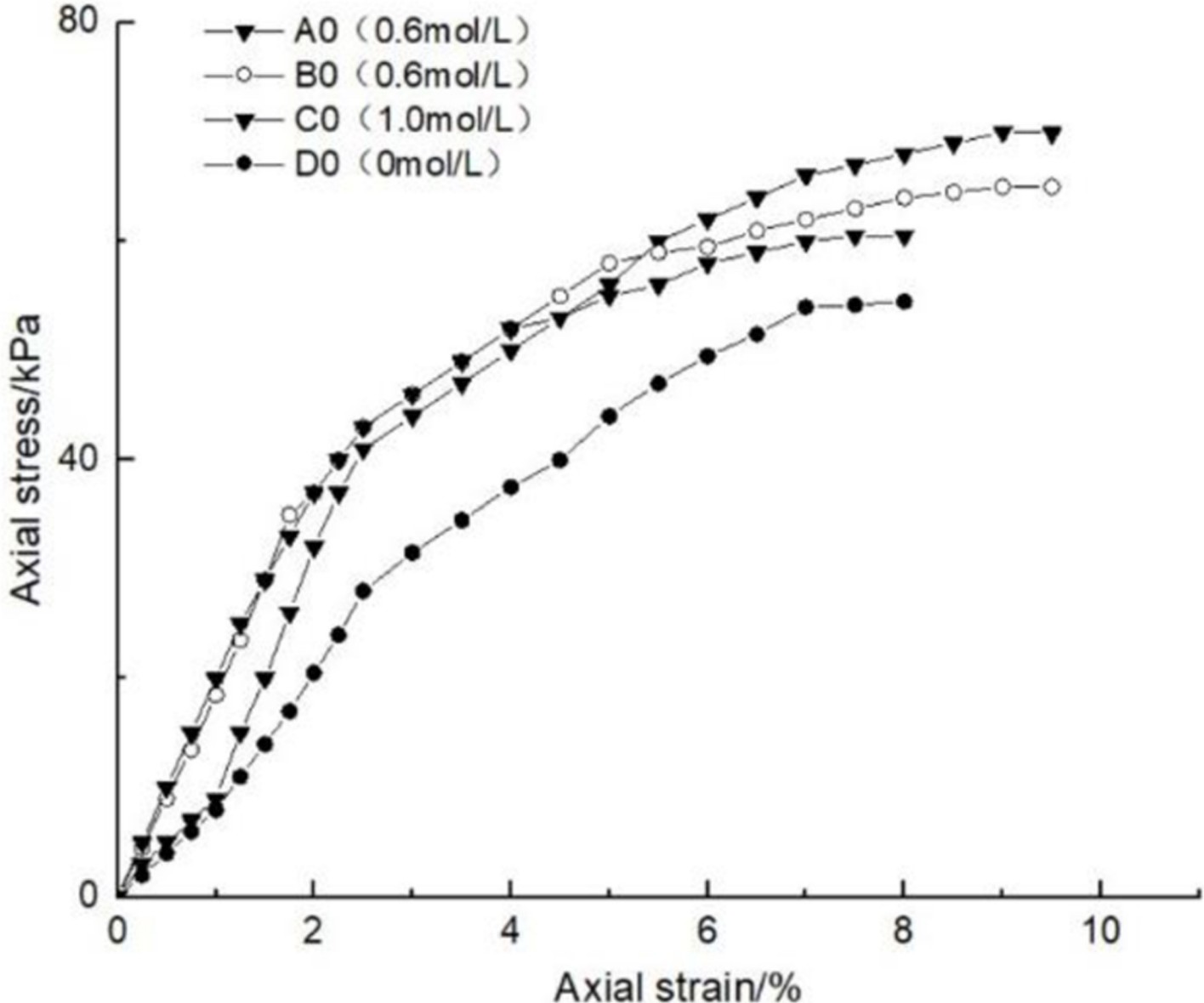

**Fig 13. Strength curve of soil with different cementing solution concentrations and a glutinous rice paste content of 0%.**

was analyzed via SEM (Zeiss sigma 500, Germany) after reducing the samples to a block size of less than 1 cm³.Images were obtained at magnifications of 2000x and 5000x, and the scanning range was -3˚ to 70˚ of the 5-axis excellent center. The changes in the morphologies and microstructures of the products were observed and the changes in the soil structure caused by the various factors were analyzed [19].

## 5.1 Characteristics of the product

Sample D0 is a silty clay sample, and it can be seen from the micro-morphology map that its surface is relatively smooth, without calcium carbonate precipitation and with flaky matured glutinous rice paste (Fig 18). The micro-morphology of sample A0 (with bacteria liquid, without

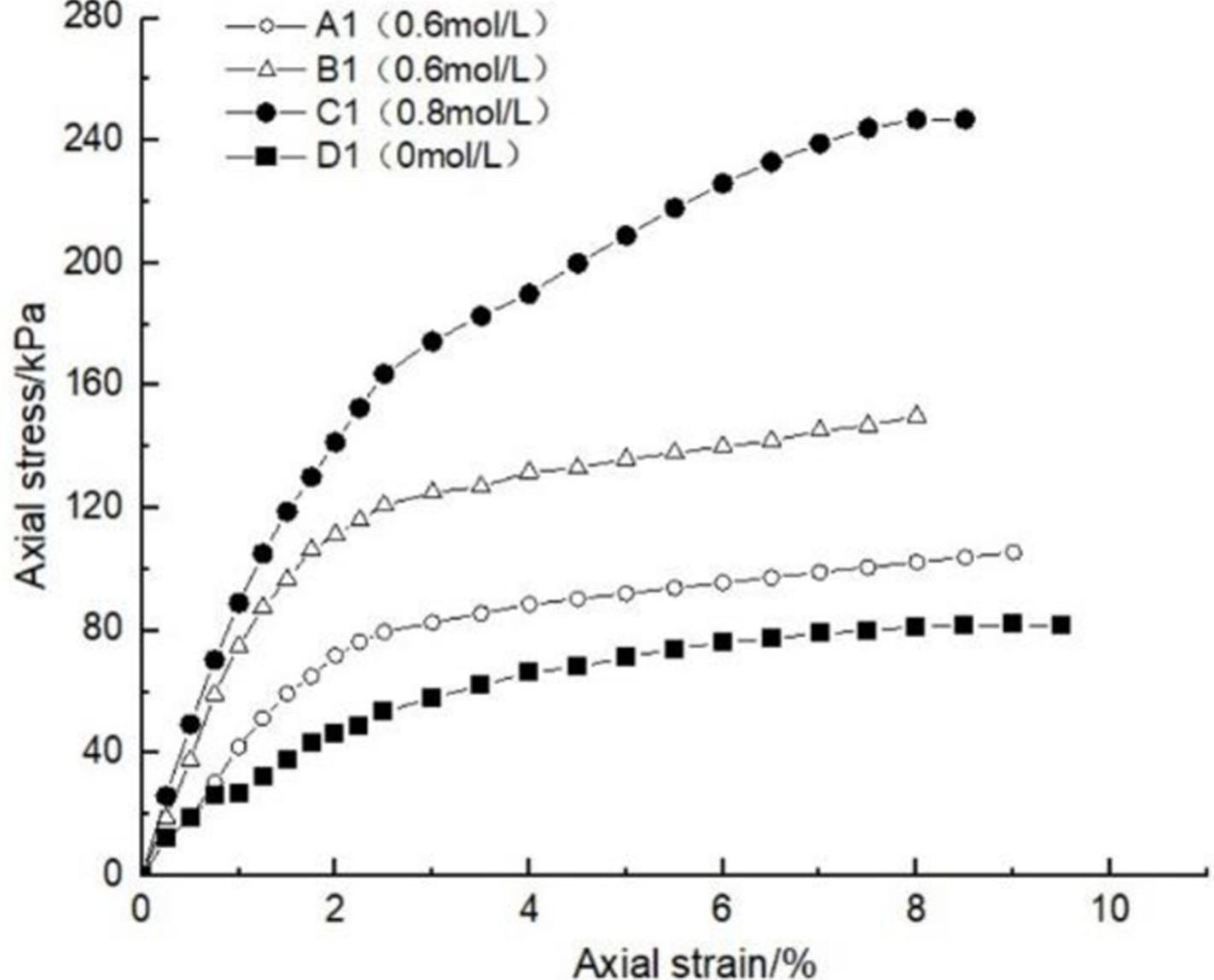

**Fig 14. Strength curve of soil with different cementing solution concentrations and a glutinous rice paste content of 3%.**

glutinous rice paste) is shown in Fig 19. It can be seen that obvious crystals have formed on the soil surfaces, medium crystals bond the soil particles, and coarse crystals fill the pores between the particles. Samples A1 and A2 contain 5% glutinous rice paste. It can be seen that a large number of crystals have formed on the surface of these soils, and there are many fine particles attached to the surfaces of the samples. Among them, there are a large number of fine particles attached to the soil surfaces of sample A2, and it is preliminarily concluded that the existence of matured glutinous rice paste caused the calcium carbonate precipitate generated via the MICP reaction more detailed, which confirms the test results of Yang Fuwei et al. [11].

It was found that there are more light-colored flakes of cooled glutinous rice paste attached to the soil surfaces and pores in the samples with matured glutinous rice paste. As is shown in Fig 19, a large number of precipitate particles formed on the surfaces of these substances after

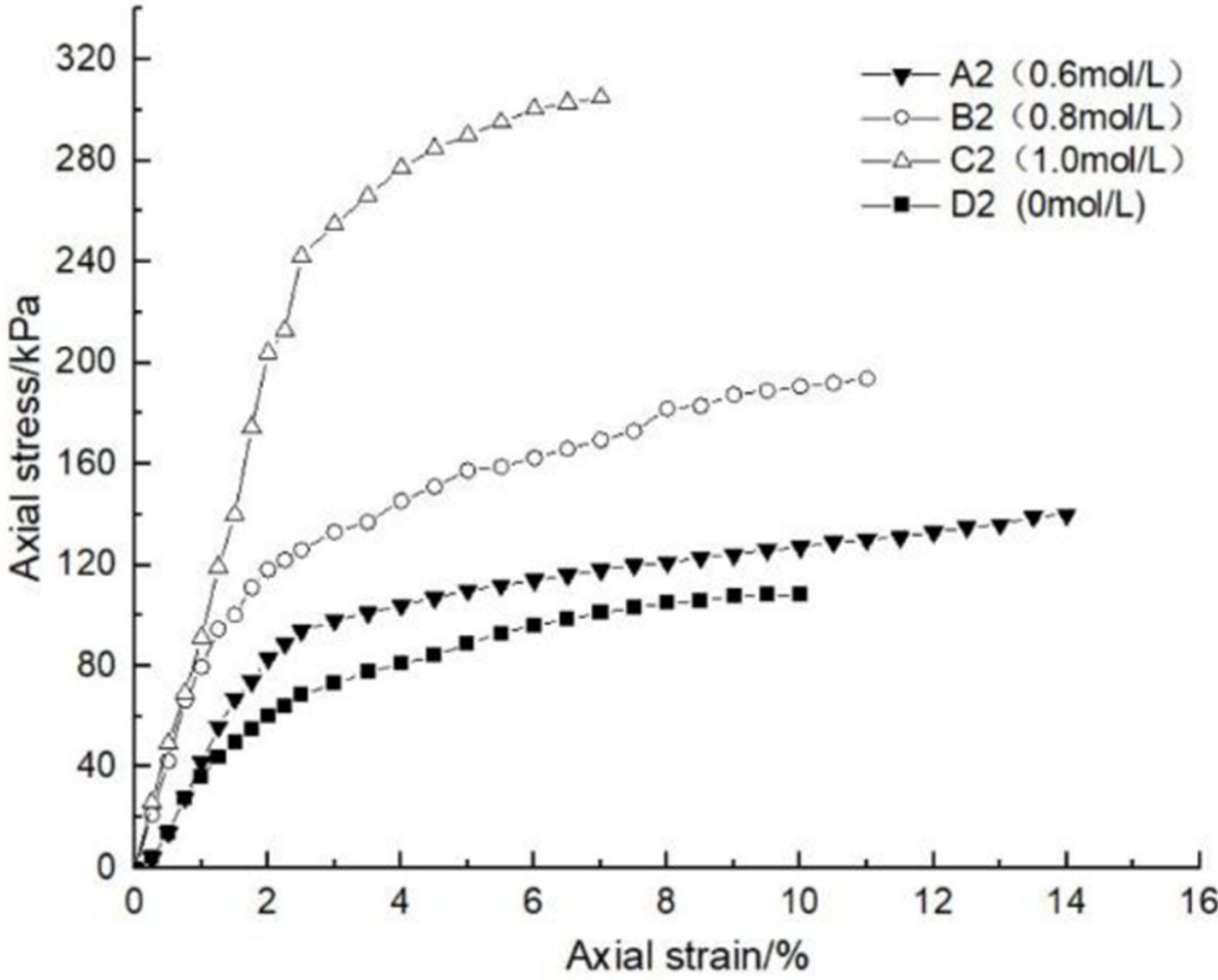

**Fig 15. Strength curve of soil with different cementing solution concentrations and a glutinous rice paste content of 5%.**

cooling, and a large number of precipitate particles are present in the pores of the samples. These substances form a bridge after cooling and hardening. The pores of the soil are filled by a large amount of calcium carbonate precipitate attached to the soil, which indirectly enhances the strength of the soil.

## 5.2 Pore characteristics

As can be seen from the SEM image of sample D2 (with cooked glutinous rice paste, without bacteria solution), there are more pores in the soil. The cooled glutinous rice paste is distributed in the pores of the soil, and because the matured glutinous rice paste and silty clay are both viscous, they were easily integrated. Therefore, simply adding matured glutinous rice paste to plain soil cannot provide additional strength after cooling.

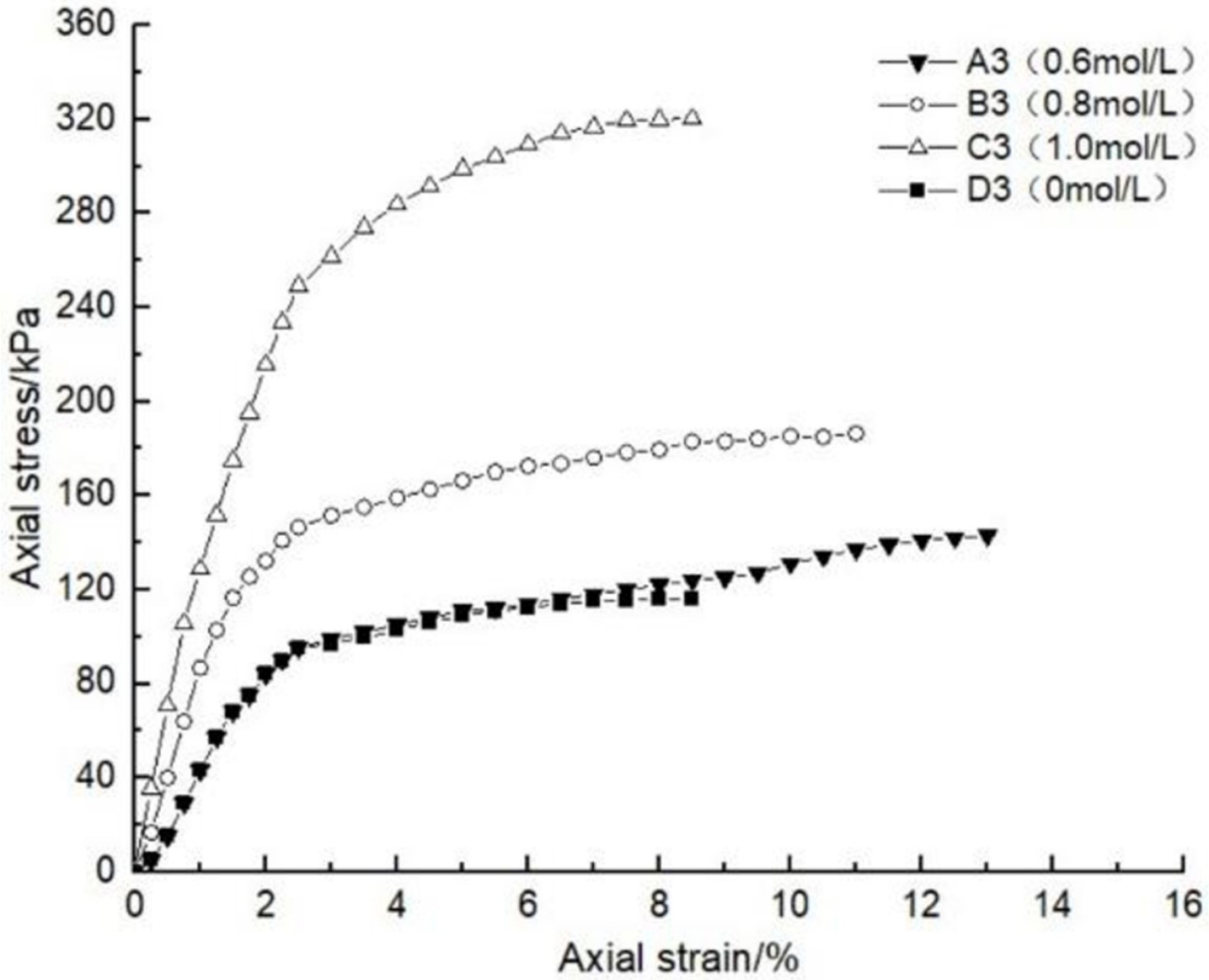

**Fig 16. Strength curve of soil with different cementing solution concentrations and a glutinous rice paste content of 7%.**

As can be seen from the SEM image(Fig 20) of sample A0 (with bacteria solution, without glutinous rice paste), a large amount of calcium carbonate precipitate formed during the MICP treatment. However, the amount of precipitate is not sufficient to fill the larger pores between the particles, and much of the precipitate is only attached to one side of the pores. Two unilateral precipitates cannot be connected, and this type of precipitation distribution is invalid precipitation. As a result, there are many pores in the soil, the structure is loose, and the compactness is poor. The strength of the sample is not high, and its unconfined compressive strength is low. This proves that the effect of a single MICP technique for treating porous soil is not ideal. Therefore, it is of great significance to study the combined effect of the filling medium and microbial solidification on the strength of the soil.

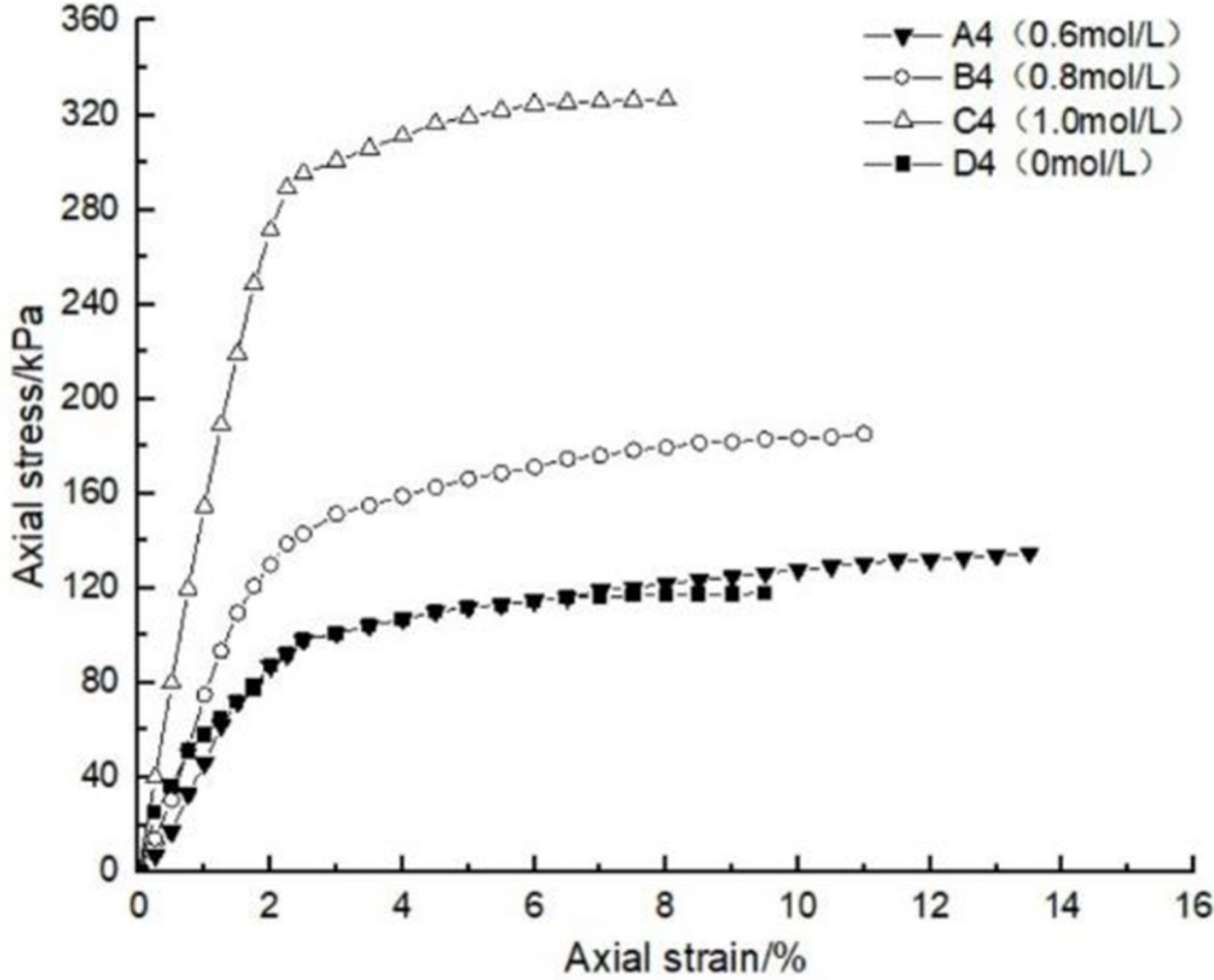

**Fig 17. Strength curve of soil with different cementing solution concentrations and a glutinous rice paste content of 10%.**

By comparing the SEM images of samples A1 and A2 (containing bacteria liquid and matured glutinous rice paste), it can be observed that the compactness of the soil increases from A1 to A2. That is, the pores between the soil particles are smaller, and the existence of cooked glutinous rice paste causes more calcium carbonate to be generated (Fig 21). The pores are filled with cooked glutinous rice paste and calcium carbonate, which indicates that the strength of the sample is increasing. This is consistent with the conclusion based on the unconfined compressive strength tests (see Section 3). Based on the experiments, we can infer that when the cooked glutinous rice paste content is 5%, as the cementing solution concentration increases, more and more coarse and fine crystals are formed in the soil. The precipitate particles not only attach to the soil surfaces, but a considerable proportion combine with the cooled flaky matured glutinous rice paste to fill the pores, which is called effective precipitation [2].

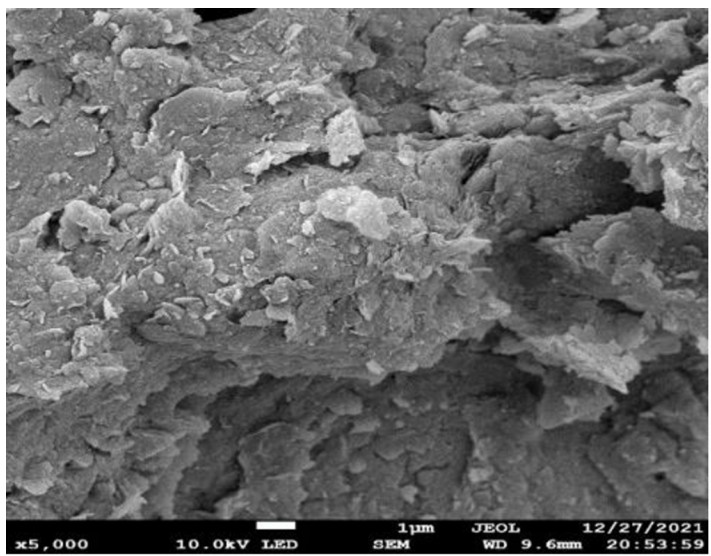
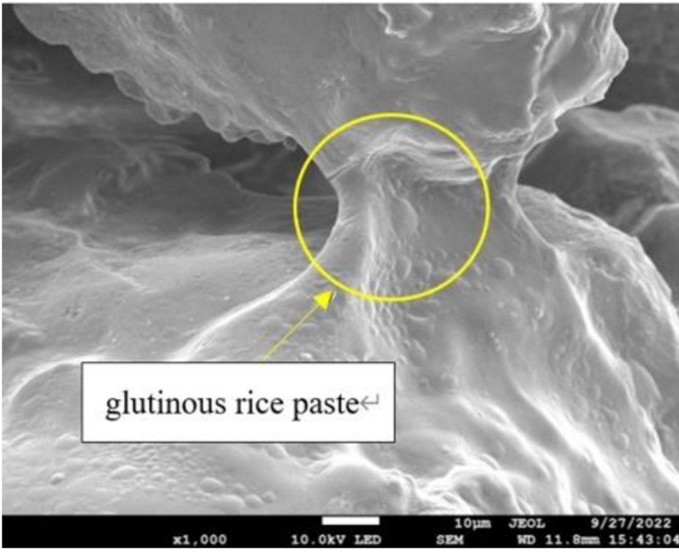

a.

b.

**Fig 18. Silty clay sample.** a. No cooked glutinous rice paste. b. Containing cooked glutinous rice paste.

## 6 Results and discussion

Based on the microbial solidification tests, strength tests, and SEM analysis of the soil samples with different dosages of matured glutinous rice paste and different concentrations of cementing solution, the following conclusions were obtained.

1. After adding common cementitious material (cured glutinous rice paste) to silty clay, it was found that the strength of the sample was improved, and the strength was up to 100% higher than that of the plain soil sample. When the concentration of the cementing solution was 0.6 mol/L, the maximum strength was only 70 kPa. When a common cementitious material (cured glutinous rice paste) and microorganisms were used to solidify the sample, the effect was the best. For the same cementitious solution concentration, the soil strength

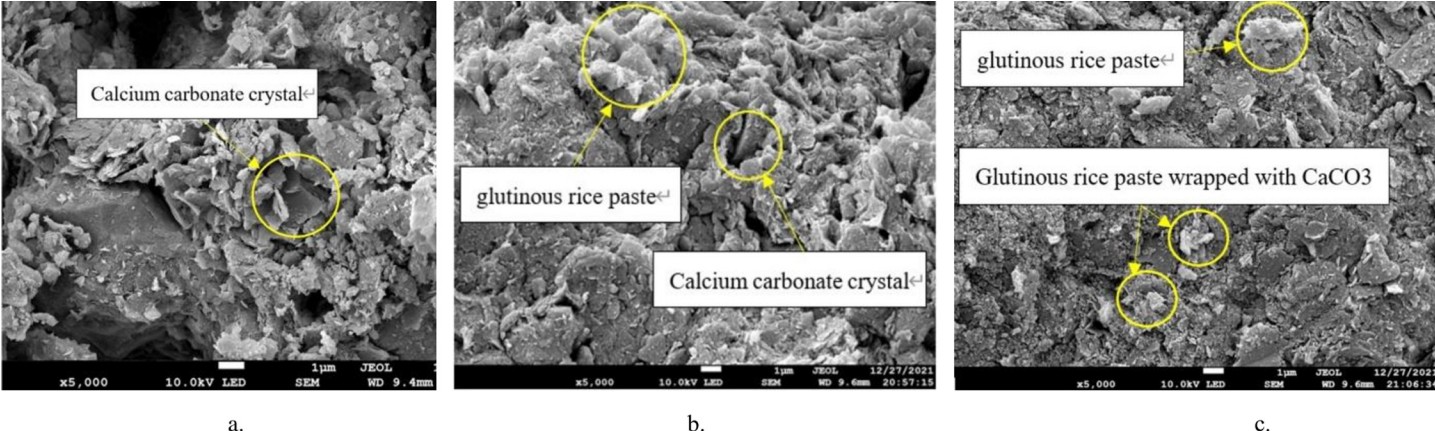

a.

b.

c.

**Fig 19. SEM images of samples with different concentrations of cooked glutinous rice paste and a cement content of 0.6 mol/L.** a. sample AO. b. sample A1. c. sample A2.

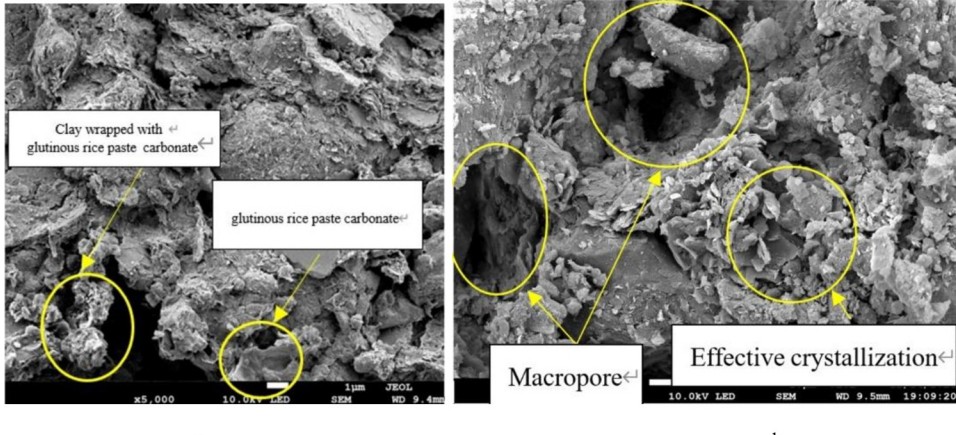

**Fig 20. SEM images of single variable silty clay.** A. With cooked glutinous rice paste added. B. With bacterial liquid added.

increased by at least 40% and at most 398.6%, and the effect was the best when the cured glutinous rice paste content was 5%.

2. The cured glutinous rice paste, as an additive, played three roles in the entire microbial curing process. First, it provided energy for the microorganisms, enhanced the microbial activity, and promoted the production of urease. The test results show that the urease activity of the reaction solution with cooked glutinous rice pulp added increased by at least 30%, which indirectly improved the utilization rate of the cementation solution, thus promoting the formation of precipitates and improving the strength of the test sample. Therefore, when the concentration of the cementation solution was ≤1 mol/L, the higher the concentration was, the higher the strength of the sample was, and the greater the strength increase was. Second, ripened glutinous rice paste has a high strength after cooling, and it can directly change the structure of the sample and enhance the strength of the sample after mixing with clay. Third, glutinous rice paste makes the precipitate particles produced by the MICP reaction finer, so they more easily combine with the cured glutinous rice paste

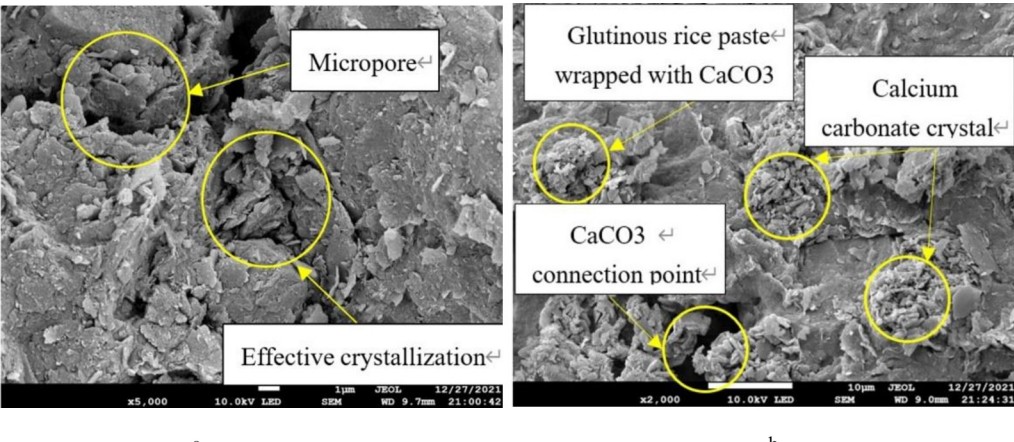

**Fig 21. SEM images of samples with different concentrations of cooked glutinous rice paste and a cement content of 0.6 mol/L.** a. sample A1. b. sample A2.

after cooling. Thus, they can fill the pores in the soil as a whole, which enhances the compactness of the soil and improves the strength of the sample.

## 7 Conclusions

The combination of ripened glutinous rice slurry and microbes for curing soil can achieve a significant increase in the soil strength, which is more economical and environmentally friendly than ordinary MICP, and the strength of the finally solidified soil is also higher. However, one limitation of this study is that only a small laboratory experiment was used for the testing, and a large range of engineering soils has yet to be tested. In order to achieve engineering applications of ripened glutinous rice slurry and microbe for curing soil, this step is indispensable. In addition, no tests have been carried out on sand and other soils, and therefore, the universality of the test results has not been confirmed. Based on this, we suggest that researchers should carry out large-scale soil engineering tests and tests on more soil types. After more test results are obtained, MICP solidified soil is expected to achieve engineering.

## Supporting information

**S1 Data.**
(ZIP)

## Author Contributions

**Data curation:** Qian Chen.

**Methodology:** Qizhi Hu.

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
