## [Decision Letter · Decision Letter 0]

16 Jan 2023

PONE-D-22-31388EXPERIMENTAL STUDY OF SYNERGISTIC REINFORCEMENT OF SILTY CLAY WITH GLUTINOUS RICE PASTE AND MICPPLOS ONE

Dear Dr. Hu,

Thank you for submitting your manuscript to PLOS ONE. After careful consideration, we feel that it has merit but does not fully meet PLOS ONE’s publication criteria as it currently stands. Therefore, we invite you to submit a revised version of the manuscript that addresses the points raised during the review process.

We look forward to receiving your revised manuscript.

Kind regards,

Nasser A. M. Barakat

Academic Editor

PLOS ONE

Journal Requirements:

“The author(Hu Qizhi) received no specific funding for this work.”

5. Please ensure that you refer to Figure 18, 20, 22, 23 and 24 in your text as, if accepted, production will need this reference to link the reader to the figure.

Reviewers' comments:

Reviewer's Responses to Questions

**Comments to the Author**

1. Is the manuscript technically sound, and do the data support the conclusions?

Reviewer #1: Yes

Reviewer #2: Yes

2. Has the statistical analysis been performed appropriately and rigorously? 

Reviewer #1: Yes

Reviewer #2: N/A

3. Have the authors made all data underlying the findings in their manuscript fully available?

Reviewer #1: Yes

Reviewer #2: Yes

4. Is the manuscript presented in an intelligible fashion and written in standard English?

Reviewer #1: Yes

Reviewer #2: No

5. Review Comments to the Author

Reviewer #1: The novelty of this study is not clear. Please highlight the strength of this study and the limitations in the last paragraph of the introduction.

In the Materials and Methods section, the methods used are not detailed enough. Please support your explanation with standards and photos.

This paper studies synergistic reinforcement of silty clay through experiments. What are the key scientific issues of this study? What laws can be obtained through this study, and what is the guiding significance for subsequent related research?

The pictures are not professional enough (Fig.3, Fig.4, Fig.6 et.) to express the key information to be highlighted. It is recommended to modify the pictures to emphasize the key points.

In the introduction part, the references cited in the literature review are numbered, but there is no reference number in the list of references at the end of the article.

Reviewer #2: The work submitted to the PLOS ONE journal entitled as “EXPERIMENTAL STUDY OF SYNERGISTIC REINFORCEMENT OF SILTY CLAY WITH GLUTINOUS RICE PASTE AND MICP” is reviewed.

This study presents An MICP solidification test of silty clay carried out by adding different concentrations of aged glutinous rice slurry and cementing liquid, and unconfined compressive strength tests and scanning electron microscope analysis of the solidified samples were carried out. The reviewer believes this research paper could be an interesting to geological and civil engineering research community and those who are interested in rock mechanics.

In general, paper is well structured, and the data is well analyzed and requires minor revision to be evaluated. I am suggesting the manuscript to be accepted for publication from the PLOS ONE however, if the authors are willing to perform major improvements / corrections on the submitted work.

Here are the major improvements / corrections I suggest authors to review:

• Some numerical results should be introduced to the abstract.

• At the last paragraph of the introduction section authors should mention about the novelty of the research.

• Please indicate the exact location of the specimen collection area.

• Authors should include further physical and chemical properties of the soil. Such as grain size distribution, specific gravity, OMC/MDD, chemical compositions etc. While mentioning those properties authors should include the standards used to obtained respective parameters.

• Figure 4 has two images please notate them and then write their description to caption.

• Authors stated that “final microbial activity was 0.312 ms/(cm·min-1), which met the requirements.” Please indicate how the requirement was set?

• Authors should justify how they have decided to come up with presented testing schema.

• Authors stated that they have prepared their samples at optimum moisture content. Is that means that they have prepared the samples at maximum dry density? If yes authors should indicate that. Also, authors should explain how that would affect the pore space of samples and effect the MICP process.

• It can be seen from the Figure 7 that authors didn’t use half sphere or full sphere between the sample and load ring. Authors should mention the effect of such approach to test results.

• It would be more technical term to use load ring instead of dynamometer.

• Instead of unconfined compression test data diagram please use stress-strain diagrams.

• Authors stated that “It was found that the strength curves of the soil with different concentrations of glutinous rice paste almost all exhibited the phenomenon that the growth rates of the early parts of the curves were the same, and the difference in the rates gradually widened in the middle part of the curve.” It is essential to use terms such as elastic range, plastic range, elastic modulus to sound more technical.

• Please mention how the SEM samples are prepared. Are they coated with carbon, gold etc.

• What do you mean by “pure vegetarian soil sample” there is no such term. The article needs extensive language editing by a native speaker in who is confident with technical terms.

• When presenting SEM images, the magnification on each image should be same to satisfactorily compare the occurrences.

• Date should be seen on Figure 17. And instead of presenting Figure 17 -21 and 22 – 25 SEM images separately please join them on a single image so that it will be easier for readers to compare the occurrences.

• General Comments – Revise the keywords according to journal guidelines.

• General comment – there is complexity on the purpose of this particular research. It should be uniquely stated what the aim is then the importance should be appreciated by those who are interested in this paper.

• General Comments – Conclusions sections should be re-arranged as Conclusion and Recommendations. In this section limitations and recommendations of this study should be listed.

• General Comments – There are some of grammatical mistakes and drawbacks in the manuscript, Please improve the English and try to present a concise expression.

• General comment – References section should be reviewed as few references are not according to the journal guidelines.

6. PLOS authors have the option to publish the peer review history of their article (what does this mean?). If published, this will include your full peer review and any attached files.

Reviewer #1: No

Reviewer #2: **Yes: **Abdullah Ekinci

---

## [Author Response · Author response to Decision Letter 0]

22 Feb 2023

Reviewer #1: 

The novelty of this study is not clear. Please highlight the strength of this study and the limitations in the last paragraph of the introduction.

This test combines the research results of traditional MICP to solidify soil, and introduces different amounts of ripened glutinous rice slurry and different concentrations of cement. The advantage of the research is that it combines the process of ripened glutinous rice slurry and microorganism to solidify soil. The cooled ripened glutinous rice slurry itself has high strength. If it is mixed with soil, the strength of the mixed soil will be significantly improved, Secondly, the cooked glutinous rice pulp can produce starch branch chain after being decomposed by microorganisms, providing energy for the survival of microorganisms, so that microorganisms can produce more urease. The quality of MICP research results is fundamentally determined by the amount of urease produced by microorganisms. Because urease can decompose urea to produce CO32 -, the amount of CO32 - determines the final amount of CaCO3 generated, and ultimately determines the strength value of soil after solidification. Based on this, Through the macroscopic strength test and microscopic electron microscope scanning test on the prepared and cured samples, the test results are analyzed to explore how the two variables interact to influence the mechanism of MICP curing soil, and provide more theoretical basis for the combined application of additives and MICP technology to soil solidification. However, this study only tests silty clay, and has not yet tested other soil bodies such as sand, loess, and other soil bodies, Whether the ripened glutinous rice slurry will react better with other soils can not be determined now. The author will continue to conduct experimental research on other soils in the follow-up study.

In the Materials and Methods section, the methods used are not detailed enough. Please support your explanation with standards and photos.

Test method is added，please see Section III of Chapter II for details.

This paper studies synergistic reinforcement of silty clay through experiments. What are the key scientific issues of this study? What laws can be obtained through this study, and what is the guiding significance for subsequent related research?

The key scientific issues include how to prepare the bacterial solution and expand the culture in the early stage, how to prepare the samples required for the test in the middle stage, including the preparation of unconfined compressive strength and scanning electron microscope samples, and finally how to process the data. In the early stage, the bacterial solution was prepared by using the instruments in the laboratory. See Section 2-1 for the details. In the later stage, the sample preparation was in Section 3-2 and the first paragraph of Chapter 5. Finally, the data processing was fitted with Origin. Through this study, we found that curing soil with cooked glutinous rice slurry and microorganism can achieve a significant increase in soil strength, which is more economical and environmentally friendly than ordinary MICP, and the final solidified soil strength is also higher, which provides a new idea for the follow-up scientific researchers to use additives and microorganism to jointly solidify soil.

The pictures are not professional enough (Fig.3, Fig.4, Fig.6 et.) to express the key information to be highlighted. It is recommended to modify the pictures to emphasize the key points.

Figure 3, Figure 4 and Figure 6 have been changed to Figure 4, Figure 5, Figure 7 because of the addition of a picture; Figure 4a shows the water mixed with glutinous rice flow, which is mushy after being heated, stirred and cooled, with poor fluidity and high viscosity. After being taken out, it is placed in a beaker. This is the additive to be used in the next test, as shown in Figure 4b; Figure 5 shows the urease activity test. The left side is the test instrument conductivity meter (FE-38), and the right side is the reaction solution under different variables, where A is the mixed solution of bacterial solution and urea, and B is the mixed solution after adding cooked glutinous rice pulp, bacterial solution and urea (only one group of samples is shown in the text); Figure 7 is the prepared sample, which is now replaced by a clearer picture. The marks on each sample in the picture correspond to the samples with different variables in the test plan. For example, "10N1J" is the sample numbered C4 in Table 2 (the content of cooked glutinous rice paste in the sample is 10%, and the concentration of cement is 1mol/L).

In the introduction part, the references cited in the literature review are numbered, but there is no reference number in the list of references at the end of the article.

References at the end of the article have been numbered.

Reviewer #2: 

The work submitted to the PLOS ONE journal entitled as “EXPERIMENTAL STUDY OF SYNERGISTIC REINFORCEMENT OF SILTY CLAY WITH GLUTINOUS RICE PASTE AND MICP” is reviewed.

This study presents An MICP solidification test of silty clay carried out by adding different concentrations of aged glutinous rice slurry and cementing liquid, and unconfined compressive strength tests and scanning electron microscope analysis of the solidified samples were carried out. The reviewer believes this research paper could be an interesting to geological and civil engineering research community and those who are interested in rock mechanics.

In general, paper is well structured, and the data is well analyzed and requires minor revision to be evaluated. I am suggesting the manuscript to be accepted for publication from the PLOS ONE however, if the authors are willing to perform major improvements / corrections on the submitted work.

Here are the major improvements / corrections I suggest authors to review:

• Some numerical results should be introduced to the abstract.

The change of strength value after curing test is introduced into the abstract.

• At the last paragraph of the introduction section authors should mention about the novelty of the research.

The last paragraph of the introduction has been re-edited, This test combines the research results of traditional MICP to solidify soil, and introduces different amounts of ripened glutinous rice slurry and different concentrations of cement. The advantage of the research is that it combines the process of ripened glutinous rice slurry and microorganism to solidify soil. The cooled ripened glutinous rice slurry itself has high strength. If it is mixed with soil, the strength of the mixed soil will be significantly improved, Secondly, the cooked glutinous rice pulp can produce starch branch chain after being decomposed by microorganisms, providing energy for the survival of microorganisms, so that microorganisms can produce more urease. The quality of MICP research results is fundamentally determined by the amount of urease produced by microorganisms. Because urease can decompose urea to produce CO32 -, the amount of CO32 - determines the final amount of CaCO3 generated, and ultimately determines the strength value of soil after solidification. Based on this, Through the macroscopic strength test and microscopic electron microscope scanning test on the prepared and cured samples, the test results are analyzed to explore how the two variables interact to influence the mechanism of MICP curing soil, and provide more theoretical basis for the combined application of additives and MICP technology to soil solidification.

• Please indicate the exact location of the specimen collection area.

The collection area is located in the project department of the fifth primary school of Optics Valley, Hongshan District, Wuhan, Hubei Province, China. The project has been completed.

• Authors should include further physical and chemical properties of the soil. Such as grain size distribution, specific gravity, OMC/MDD, chemical compositions etc. While mentioning those properties authors should include the standards used to obtained respective parameters.

According to the data provided by the project department, the particle size distribution curve of the soil sample is prepared.

• Figure 4 has two images please notate them and then write their description to caption.

Figure 4 has now been changed to Figure 5, and the two pictures in Figure 5 are FE-3 conductivity instrument (left) and reaction solution (right).

• Authors stated that “final microbial activity was 0.312 ms/(cm·min-1), which met the requirements.” Please indicate how the requirement was set?

According to the article titled "Microbial CaCO3 precision for the production of microorganism" published by Victoria S. Whitfin on Murdoch University, according to the empirical formula, the urease activity value can be determined by the conductivity change rate of the bacterial solution in the first five minutes. In this study, the author took the final average value after many tests to draw the conductivity curve, It is calculated that the conductivity change rate of bacterial solution in the first five minutes is 0.312 mS/(cm · min-1). At the same time, urease, as an enzyme that catalyzes the hydrolysis of urea to ammonia and carbon dioxide, its activity represents the ability to hydrolyze urea. In the research direction of microorganisms, there are many microorganisms with urease activity of 0 ms/(cm · min-1). In MICP research, Only when the urease activity is greater than 0 ms/(cm · min-1) can it play an effective role in the process of soil consolidation. In order to facilitate readers' understanding, the author has replaced the original conductivity diagram with urease activity diagram.

• Authors should justify how they have decided to come up with presented testing schema.

There are three test variables in this study, which are the concentration of cement, the amount of cooked glutinous rice paste and the amount of bacterial liquid. The purpose is to explore the following three points:

1. Verify the influence of cement concentration on MICP process;

2. Verify the effect of the concentration of cooked glutinous rice slurry on the microbial solidification of soil;

3. Verify whether the increase in soil strength is completely caused by the cooked glutinous rice slurry, and set a control group with a bacterial liquid content of 0ml.

The control variable method is used to formulate the test plan suitable for the purpose of this experiment, as shown in Table 2.

• Authors stated that they have prepared their samples at optimum moisture content. Is that means that they have prepared the samples at maximum dry density? If yes authors should indicate that. Also, authors should explain how that would affect the pore space of samples and effect the MICP process.

The samples used in this test are prepared at the maximum dry density. The soil porosity under the maximum dry density will not affect the solidification process of MICP, because the microorganism used in this study is Bacillus, and the diameter of the strain is less than 10 μ m. Calcium carbonate crystals generated during curing are less than 5 μ m. During the MICP process, the calcium carbonate crystal will gradually form a larger diameter calcium carbonate precipitate with Bacillus as the nucleation point until it fills the whole pore.

• It can be seen from the Figure 7 that authors didn’t use half sphere or full sphere between the sample and load ring. Authors should mention the effect of such approach to test results.

All parts of YYW-2 instrument are shown in Figure 7.

• It would be more technical term to use load ring instead of dynamometer.

Has been changed.

• Instead of unconfined compression test data diagram please use stress-strain diagrams.

Has been changed.

• Authors stated that “It was found that the strength curves of the soil with different concentrations of glutinous rice paste almost all exhibited the phenomenon that the growth rates of the early parts of the curves were the same, and the difference in the rates gradually widened in the middle part of the curve.” It is essential to use terms such as elastic range, plastic range, elastic modulus to sound more technical.

The curvature of the curve is almost the same at the initial stage, which means that the sample is at the elastic stage, and the strength of the soil itself is sufficient to resist external forces; The rate difference in the middle part of the curve gradually expands, which means that the internal part of the sample starts to break and gradually enters the elastic-plastic stage, and the strength of different samples is different, so the curve curvature at the elastic-plastic stage is different.

• Please mention how the SEM samples are prepared. Are they coated with carbon, gold etc.

The preparation method of SEM samples is described in the first paragraph of Chapter 5, in which the samples are coated with gold.

• What do you mean by “pure vegetarian soil sample” there is no such term. The article needs extensive language editing by a native speaker in who is confident with technical terms.

"Vegetarian soil sample" is a language expression error and has been changed (this article has been polished by Accdon-LetPub Editor). 

• When presenting SEM images, the magnification on each image should be same to satisfactorily compare the occurrences.

The multiples selected in the article are all 5000 times, but there are two different multiples, the 18th and the 23rd. The 23rd author wants to show the SEM image of the cooked glutinous rice paste. Too high multiples will make the whole of the cooked glutinous rice paste not be fully displayed. The image magnification of the 23rd is 2000 times, because the author wants the reader to see that a wide range of pores are filled by calcium carbonate precipitation, Other images with different multiples have been replaced with images with the same multiples.

• Date should be seen on Figure 17. And instead of presenting Figure 17 -21 and 22 – 25 SEM images separately please join them on a single image so that it will be easier for readers to compare the occurrences.

The picture classification has been merged.

• General Comments – Revise the keywords according to journal guidelines.

Has been changed.

• General comment – there is complexity on the purpose of this particular research. It should be uniquely stated what the aim is then the importance should be appreciated by those who are interested in this paper.

At present, the MICP studied is based on the grouting method, and the grouting method uses a large amount of bacterial fluid, and the price of bacterial species is expensive ($120 per gram), so the cost of the grouting method is high. The purpose of this study is to explore the effect of microbial combined additives (such as cooked glutinous rice slurry) on soil solidification. Once the test results are good, then the test results can be used in practice, This means that a small amount of bacterial solution can significantly increase the strength of soil mass, which will greatly reduce the test cost and is expected to realize the practice of MICP.

• General Comments – Conclusions sections should be re-arranged as Conclusion and Recommendations. In this section limitations and recommendations of this study should be listed.

The last section has been changed to conclusions and recommendations, and the limitations and suggestions of the study have been emphasized in the recommendations.

• General Comments – There are some of grammatical mistakes and drawbacks in the manuscript, Please improve the English and try to present a concise expression.

This article has been edited and polished by letpub. The grammatical error in the article may be that the polishing editor has not fully understood the meaning of the article and has now been retouched.

• General comment – References section should be reviewed as few references are not according to the journal guidelines.

Has been changed.

---

## [Decision Letter · Decision Letter 1]

5 Apr 2023

Experimental study of synergistic reinforcement of silty clay with glutinous rice paste and MICP

PONE-D-22-31388R1

Dear Dr. Hu,

We’re pleased to inform you that your manuscript has been judged scientifically suitable for publication and will be formally accepted for publication once it meets all outstanding technical requirements.

Kind regards,

Nasser A. M. Barakat

Academic Editor

PLOS ONE

Additional Editor Comments (optional):

Reviewers' comments:

Reviewer's Responses to Questions

**Comments to the Author**

1. If the authors have adequately addressed your comments raised in a previous round of review and you feel that this manuscript is now acceptable for publication, you may indicate that here to bypass the “Comments to the Author” section, enter your conflict of interest statement in the “Confidential to Editor” section, and submit your "Accept" recommendation.

Reviewer #2: All comments have been addressed

2. Is the manuscript technically sound, and do the data support the conclusions?

Reviewer #2: Yes

3. Has the statistical analysis been performed appropriately and rigorously? 

Reviewer #2: Yes

4. Have the authors made all data underlying the findings in their manuscript fully available?

Reviewer #2: Yes

5. Is the manuscript presented in an intelligible fashion and written in standard English?

Reviewer #2: Yes

6. Review Comments to the Author

Reviewer #2: The work submitted to the PLOS ONE journal entitled as “EXPERIMENTAL STUDY OF SYNERGISTIC REINFORCEMENT OF SILTY CLAY WITH GLUTINOUS RICE PASTE AND MICP” is re-reviewed.

In general, the authors have successfully answered and reflected most of the concerns arise in the previous round of review. The article can now be accepted as is.

7. PLOS authors have the option to publish the peer review history of their article (what does this mean?). If published, this will include your full peer review and any attached files.

Reviewer #2: **Yes: **Abdullah Ekinci

---

## [Editor Report · Acceptance letter]

11 Apr 2023

PONE-D-22-31388R1 

Experimental study of synergistic reinforcement of silty clay with glutinous rice paste and MICP 

Dear Dr. Hu:

I'm pleased to inform you that your manuscript has been deemed suitable for publication in PLOS ONE. Congratulations! Your manuscript is now with our production department. 

Kind regards, 

on behalf of

Dr. Nasser A. M. Barakat 

Academic Editor

PLOS ONE